# QCM Sensor Arrays, Electroanalytical Techniques and NIR Spectroscopy Coupled to Multivariate Analysis for Quality Assessment of Food Products, Raw Materials, Ingredients and Foodborne Pathogen Detection: Challenges and Breakthroughs [note 1]

**DOI:** 10.3390/s20236982

**Published:** 2020-12-07

**Authors:** David K. Bwambok, Noureen Siraj, Samantha Macchi, Nathaniel E. Larm, Gary A. Baker, Rocío L. Pérez, Caitlan E. Ayala, Charuksha Walgama, David Pollard, Jason D. Rodriguez, Souvik Banerjee, Brianda Elzey, Isiah M. Warner, Sayo O. Fakayode

**Affiliations:** 1Chemistry and Biochemistry, California State University San Marcos, 333 S. Twin Oaks Valley Rd, San Marcos, CA 92096, USA; dbwambok@csusm.edu; 2Department of Chemistry, University of Arkansas at Little Rock, 2801 S. University Ave, Little Rock, AR 72204, USA; nxsiraj@ualr.edu (N.S.); spmacchi@ualr.edu (S.M.); 3Department of Chemistry, University of Missouri, 601 S. College Avenue, Columbia, MO 65211, USA; nelqmb@mail.missouri.edu (N.E.L.); bakergar@missouri.edu (G.A.B.); 4Department of Chemistry, Louisiana State University, 232 Choppin Hall, Baton Rouge, LA 70803, USA; rperez@lsu.edu (R.L.P.); ayala1@lsu.edu (C.E.A.); iwarner@lsu.edu (I.M.W.); 5Department of Physical Sciences, University of Arkansas-Fort Smith, 5210 Grand Ave, Fort Smith, AR 72913, USA; Charuksha.Walgama@uafs.edu (C.W.); Souvik.Banerjee@uafs.edu (S.B.); 6Department of Chemistry, Winston-Salem State University, 601 S. Martin Luther King Jr Dr, Winston-Salem, NC 27013, USA; pollardda@wssu.edu; 7Division of Complex Drug Analysis, Center for Drug Evaluation and Research, US Food and Drug Administration, 645 S. Newstead Ave., St. Louis, MO 63110, USA; Jason.Rodriguez@fda.hhs.gov; 8Science, Engineering, and Technology Department, Howard Community College, 10901 Little Patuxent Pkwy, Columbia, MD 21044, USA; belzey@howardcc.edu

**Keywords:** food-quality-assessment, near infrared spectroscopy, multivariate analyses, analysis quartz crystal microbalance, electroanalytical sensors

## Abstract

Quality checks, assessments, and the assurance of food products, raw materials, and food ingredients is critically important to ensure the safeguard of foods of high quality for safety and public health. Nevertheless, quality checks, assessments, and the assurance of food products along distribution and supply chains is impacted by various challenges. For instance, the development of portable, sensitive, low-cost, and robust instrumentation that is capable of real-time, accurate, and sensitive analysis, quality checks, assessments, and the assurance of food products in the field and/or in the production line in a food manufacturing industry is a major technological and analytical challenge. Other significant challenges include analytical method development, method validation strategies, and the non-availability of reference materials and/or standards for emerging food contaminants. The simplicity, portability, non-invasive, non-destructive properties, and low-cost of NIR spectrometers, make them appealing and desirable instruments of choice for rapid quality checks, assessments and assurances of food products, raw materials, and ingredients. This review article surveys literature and examines current challenges and breakthroughs in quality checks and the assessment of a variety of food products, raw materials, and ingredients. Specifically, recent technological innovations and notable advances in quartz crystal microbalances (QCM), electroanalytical techniques, and near infrared (NIR) spectroscopic instrument development in the quality assessment of selected food products, and the analysis of food raw materials and ingredients for foodborne pathogen detection between January 2019 and July 2020 are highlighted. In addition, chemometric approaches and multivariate analyses of spectral data for NIR instrumental calibration and sample analyses for quality assessments and assurances of selected food products and electrochemical methods for foodborne pathogen detection are discussed. Moreover, this review provides insight into the future trajectory of innovative technological developments in QCM, electroanalytical techniques, NIR spectroscopy, and multivariate analyses relating to general applications for the quality assessment of food products.

## 1. Introduction and Overview

Secure and sustainable access to safe, quality foods is undeniably among the most significant global challenges of the 21st century. Access to safe and quality foods is also a top global priority for food manufacturing and processing industries, regulatory agencies, public health officials, and other food stakeholders. Routine and effective quality checks, assessments, and assurances of food products, raw materials, and food ingredients is also pertinent and critically important to ensure foods of high quality to safeguard food safety and public health. Guaranteed access to safe, authentic, quality foods is only achievable via proactive, concerted, and well-coordinated efforts among food manufacturing and processing industries, public health officials, and regulatory agencies. Accordingly, agencies such as the United States Foods and Drug Administration, United States Department of Agriculture, European Commission, European Food Safety Authority, World Health Organization, and the Food and Agricultural Organization of the United Nations are working collaboratively to mitigate the sales of fraudulent, substandard, adulterated, and unsafe foods [1,2,3,4,5,6,7]. Non-governmental agencies have also played critical roles in promoting access to safe foods for millions of people globally, particularly during pandemics, natural disasters, and in emergencies.

Nevertheless, quality checks, assessments, and assurances of food products along distribution chains are impacted by various challenges. This review article includes a literature survey and a summary of general challenges and major breakthroughs in methods of quality checks and assessments of a variety of foods, food raw materials, and ingredients. The review is limited to journals, books, and reviewed articles published in English. Specifically, recent technological innovations and notable advances in methods involving quartz crystal microbalances (QCM), sensors, electroanalytical techniques, and NIR spectroscopic instrumentation for use in the quality assessment of selected foods, raw materials, and ingredients from January 2019 through July 2020 are highlighted. In addition, chemometric approaches and multivariate analyses of spectral data for NIR instrumental sample analysis and calibration for quality assessment and assurance of selected foods are discussed. Moreover, the review provides insight into the future direction of innovative technological developments in QCM, electroanalytical techniques, NIR spectroscopy, and multivariate analyses for the quality assessment of food products.

## 2. General Challenges and Breakthroughs in Quality Assessment of Food Products

The development of a strategy to promote production and ensure the intake of safe foods is an active area of research and a top global priority for the food manufacturing and processing industries, public safety officials, regulatory agencies, and other food stakeholders. Yet, food quality assessments face tremendous technological challenges involving instrumentation, analytical method development, method validation, political issues, and limited resources [8,9,10,11,12,13,14,15,16,17,18,19,20,21]. For instance, the sales of counterfeit and/or adulterated food products by cartels for illicit financial gains pose a global challenge concerning food safety and public health. Significant percentages of food products are poorly monitored and inadequately assessed for quality and safety for a variety of reasons [8,9,10,11,12,13,14,15,16,17,18,19,20,21].

The development of a portable, low-cost instrument that is capable of rapid, reliable, sensitive, accurate, and robust, real-time quality checks, assessments, and assurances of food products in the field and/or at the production line in a food manufacturing or processing industry is still a big challenge. The development of new equipment is mostly market and often profit driven. Instrument developing companies are deliberate and intentional when investing in new technology in an effort to develop instruments that are financially viable and profitable.

Chromatographic separations coupled with mass spectrometry (GC-MS and HPLC-MS) and nuclear magnetic resonance (NMR) are matured techniques and have been well developed for food quality assessments. Despite the high sensitivity and accuracy of current reference protocols, they suffer significant drawbacks that preclude their wider applicability for fast screening in quality food assessment. Both GC-MS and HPLC-MS are particularly slow, require lengthy sample preparation, extraction, cleaning-up procedures, and expensive instrumentation. Moreover, portable GC-MS and HPLC-MS techniques are not widely available, impeding their practical application for rapid on-site food quality assessment, routine analysis, and rapid field study.

The development of analytical protocols for the detection of emerging/new contaminants analysis in foods matrices remains an analytical challenge of top priority in the food processing or manufacturing industries. Another significant challenge in the method validation strategy is the non-availability of reference materials and/or standards for emerging food contaminants. The use of multi-calibration for rapid, sensitive, and accurate analyses of multicomponent analytes for quality food assessment is limited and requires a significant undertaking. Poor analytical methodologies involving selectivity, specificity, and the reliable detection of potential toxins and contaminants in food matrixes at low concentrations is problematic. Type I errors (false negative) and Type II errors (false positive) of low analyte detection is concerning and a challenge in food quality assessment. Moreover, the detection of food-borne pathogens and microbial contamination in real time along a supply chain remains a huge task of top priority for food safety and public health. Moreover, a continued decline in government research support hinders the creative innovation of new instrumental and technological method development in academia and national labs. In addition, limited resources (material, financial, and workforce), a lack of political will/power, and ineffective food quality monitoring schemes by regulatory agencies are challenges, hindering the effective assurance of sufficient food quality.

Despite the highlighted challenges, notable progress has been achieved in food quality assessment in recent years. An area of active research leading to a breakthrough in food quality assessment is the development of sensors. Recent advancements in nanotechnology and material science has facilitated the development of chemical sensors in food processing and food packaging industries [22,23]. A significant breakthrough was achieved in the development of portable electroanalytical devices, including electronic noses and electronic tongues for quality food flavorings assessment [23]. Critical roles of sensors and automation techniques in food quality assessment, including process monitoring, shelf-life investigation, freshness evaluation, and authenticity in food processing industries, have been demonstrated [23]. Emerging low-cost and reliable technology based on machine learning (ML) and artificial intelligence (AI), computer vision, biometrics, and robotics for the accurate, remote, automated assessment of the quality and consumer preference of beverage and for the assessment of food smells have resulted in major breakthroughs [24,25].

The development of DNA barcoding and DNA barcode scanners in recent decades is a significant breakthrough that has facilitated rapid food quality assessment for authenticity [26]. The use of DNA barcode scanners affords rapid detection of food fraud, adulterations, and mislabeling to ensure food quality assurance along a distribution chain [26]. DNA barcoding scanners are potable, low cost, and require no technical training to use, thereby promoting rapid scanning and authentication of food products. Importantly, DNA barcoding is high-throughput, sensitive, and with multiplexing capability. However, the use of a DNA barcode scanner for quality food assessment is very limited in developing countries. In addition, DNA barcode scanners are not readily available at the point of processing or in the field, further limiting its wide applicability in quality food assessment.

Applications of multiplex real-time polymerase chain reaction (PCR) protocols for fast screening of food products for quality assessment is a significant development. The practical utility of PCR for accurate detection of adulterated meat in mutton [27], detection of pork in meat and meat products [28], chicken adulteration [29], and the content of camel milk in adulterated milk samples was successfully demonstrated [30]. Importantly, the capability of PCR for the detection of a variety of foodborne pathogens, including *Salmonella enterica, Escherichia Coli*, and *Campylobacter jejuni* [31,32,33,34], *Staphylococcus aureus* [35], *Toxoplasma gondii* [36], and many foodborne pathogens in food samples have been recently reported. Moreover, the use of PCR for the GMO screening of food products with high specificity, good accuracy, and a low limit of detection has been demonstrated [37].

The simplicity, non-invasive, non-destructive properties of NIR and FTIR spectroscopy makes it appealing and a desired analytical method of choice for rapid quality checks, assessments and the assurance of food products, raw materials, and ingredients [38]. Detailed review of innovations in the use of NIR spectroscopy for assessment of food quality is provided in different sections of the review article. Recent technological innovations and advances have also promoted the development of a low-cost hand-held portable NIR spectrometer with improved sensitivity which has incredibly facilitated routine, in-situ, on-site detection, and rapid quality food assessment in the field in real-time. Recent technological innovations, advances, and breakthroughs in NIR, QCM, and electroanalytical instrument development is extensively covered in Section 3 of the review article. Furthermore, the combined use of NIR spectroscopy and multivariate regression analysis data processing strategies has further promoted the wider applicability of NIR for fast screening, authentication, and quality assessment of a diverse variety of foods and food products, food raw materials, and ingredients, with incredible precision, excellent accuracy, and high sensitivity. Section 4 of the review highlights innovations in the use of NIR spectroscopy and multivariate regression analysis data processing strategies for food quality assessment.

## 3. Recent Technological Innovations and Advances in NIR, QCM, and Electroanalytical Spectroscopic Instrument Development

Technological innovations and advances in development of NIR spectroscopic instruments, quartz crystal microbalance (QCM) and electroanalytical techniques is motivated by their applications for the sensitive, selective, non-invasive, and non-destructive analysis of consumer products and quality control assurance. In addition, these techniques are simple, low-cost, and portable, which enables their applications for robust, accurate, and rapid purity analysis, as well as the quality control and assurance of consumable products. As a result, there are several recent developments in these instrumental techniques. These recent innovations are highlighted in the following subsections for each of these instrumental techniques: (i) near infrared (NIR) spectroscopy, (ii) quartz crystal microbalance (QCM), and (iii) electroanalytical techniques.

### 3.1. NIR Spectroscopy

Recent advances in near infrared (NIR) spectroscopy is due to its applications for sensitive and noninvasive analysis and the capability of sample analysis in transmission or reflectance mode. In addition, NIR analysis can be performed on thin and thick samples with minimum sample preparation [38]. Importantly, NIR spectroscopy has undergone rapid transformation and growth in miniaturization and the development of low-cost, handheld devices. As a result, NIR techniques have become invaluable for testing food and beverages for industrial and consumer applications. For example, a low-cost portable NIR instrument was developed and used for the determination of nutritional parameters of pasta sauce blends. Six different nutritional parameters of a broad range of different pasta/sauce blends including energy, protein, fat, carbohydrates, sugar, and fiber were recorded by a handheld NIR spectrometer. In addition, application of chemometric approaches and calibration strategies such as partial least squares regression (PLS) to the handheld NIR spectrometer in food quality assessment have been demonstrated [39]. Moreover, the applications of a combined use of PLS regression, electronic tongue, and NIR spectroscopy for the accurate quality assessment of melon varieties, including grafted and the self-rooted types, have been demonstrated [40]. Detailed applications of chemometric approaches and calibration strategies in NIR spectroscopy for food sample analysis and food quality assessment is provided in Section 9 and Section 10 of the review article.

### 3.2. Portable NIR Sensors

Portable NIR sensors have gained increased attention due to the convenience for onsite analysis. A portable NIR sensor was developed by Lucia and coworkers and used for detection of biofilms of *Staphylococcus aureus* on surfaces [41]. Portable NIR sensors have also been used to evaluate the quality of pork [42]. The method could be used to categorize pork into the four groups based on quality that is determined by pig’s genotype and/or the feeding regimen. A handheld micro electro mechanical instrument was used to acquire the spectra by scanning live animal skin, carcass surface, fresh meat and subcutaneous fat samples [42]. In addition, portable handheld NIR instruments have found applications for space exploration in Mars. The handheld NIR spectrometers are capable of analyses of the composition of minerals and geological materials of various types and sizes in space in situ [43]. NIR spectroscopy is increasingly used as a remote sensing technique for the surface characterization of planetary objects. For example, NIR spectrometer/Supercam instrument was used onboard Mars2020Rover for the identification of a variety of mafic and altered minerals on the surface of Mars [44]. Such spectrometer development will potentially facilitate the assessment of food quality in a space station laboratory in the future.

Seng and coworkers [45] have also developed a new NIR sensing technique known as pre-dispersive NIR light sensors with five light emitting diodes (at 780, 850, 870, 910, and 940 nm) in the NIR wavelengths for pineapples’ quality assessment. In addition to the low cost, the combined use of this sensor with an artificial neural network facilitated accurate and a reliable quality assessment of intact fruits [45]. NIR sensors that are capable of quality control in a food processing and manufacturing industry was also developed during the review period. For example, Eskildsen and coworkers [46] have developed NIR spectroscopy instruments for online assessment of fat and dry matter in cheese blocks at different processing stages. Although time-resolved NIR spectroscopy is regarded as a standard technique for determination of absorbance and optical characteristics of tissues, it requires a means of differentiating the signal from the instrument from the sample measurement. To resolve this challenge, Wojtkiewicz et al. [47] have developed a new self-calibrating instrument which eliminates the influence of the instrument response function from the data [47]. The accuracy of this self-calibrating instrument was validated using phantom and in-vivo data from two different time-resolved instruments. Importantly, the recently reported approach showed that recovery of parameters in a multi-wavelength time-resolved data can be achieved without prior instrument calibration [47]. Other notable NIR innovations in instrumental methods for spectral analysis during the review period includes a dual detection channel instrument for NIR time-resolved diffuse optical spectroscopy [48], functional NIR spectroscopy (fNIRS) instruments for high-density diffuse optical tomography [49,50], and other NIR-based sensors [51,52,53,54,55].

### 3.3. Quartz Crystal Microbalance (QCM)

The detection principle in a quartz crystal microbalance (QCM) is based on the change in frequency of a quartz resonator due to a change in mass per unit area. Yu-zhi and coworkers [56] developed a portable QCM instrument based on difference frequency detection between the reference and detection crystal (between ±10 to ±30 kHz). This QCM instrument showed remarkable accuracy, precision and stability. For instance, a frequency drift of less than 0.13 Hz/min and 0.23 Hz/min, respectively, were obtained for gas phase and pure water. In addition, an accuracy of less than 0.0028% (relative error from the difference frequency) and precision of less than 0.2825% (the error difference frequency between theoretical value and measurement value) were obtained [56]. A modulated reference frequency source was generated using a direct digital synthesizer. Another advantage of this instrument is the ability to adjust the reference frequency based on conditions of the experiment. The instrumental response was good as shown by a linear relationship between the change in frequency as a function of density of sodium chloride and change in viscosity of glycerol. In addition, this instrument could be coupled with electrochemical sensors for online detection of copper during deposition. The mass response value was 0.61 ng/Hz, which represented 82% of the theoretical value [56]. Petteri and Tapani [57] and Dirri et al. [58] have also highlighted new innovations and applications in QCM sensors. The practical applications of QCM in food quality assessment is provided in Section 12 of this review article. Electroanalytical techniques including electronic nose [59], electronic tongue [60,61], and biosensors [62] have notably gained attention for quality control assessment due to their sensitivity and portability. Some of these sensors are nanomaterial-based [50,62,63,64], carbon electrode [65,66], lab-on-a-disc (LOD) [67], paper-based multiplexed [68], and electrochemical aptamer-based (E-Aptasensor) [69] sensors. The real-world applications of electrochemical biosensors for the quality assessment of food products and for the detection of foodborne pathogens are provided in Section 13 of this review.

## 4. NIR Spectroscopy in Processed Foods

The use of NIR spectroscopy for the quality assessment of processed foods has generated a lot of interest during the review period. This section of the review highlights the recent innovation and reported use of NIR spectroscopy for the quality assessment of dairy products, grains and flours, chocolate and syrups, herbs and spices, and food additives.

### 4.1. Edible Oils

Despite the similar vibrational modes of monounsaturated fatty acids (e.g., oleic acid, C18:1), NIR spectral variations have been recently exploited for quality control and the compositional determination of adulterations in extra virgin olive oil and other edible oils. In a recent report, Sohng et al. [70] utilized two-dimensional correlation analysis of temperature-dependent NIR spectral variations to discriminate fatty acid content in several mock-adulterated olive oils. In particular, they propose a rapid quality control technique for determining olive oil adulteration by employing a mild heating gradient (20–41 °C) to discern subtle temperature-dependent NIR spectral variations, accomplishing an accuracy (considering instances of true versus false positives/negatives for adulteration) of 86.4%. Likewise, the discrimination of extra virgin olive oil from adulteration oils was achieved using first-order data from room temperature NIR spectra alongside chemometric tools, particularly principle component analysis (to reveal relationships between olive oils and their common adulterants) and partial least squares discriminant modeling [71]. Impressively, the latter chemometric tool boasted 100% accuracy when approximating the geographic region of origin of over 100 Argentinean olive oils. Similar chemometric models were also employed for the discrimination of geographic origin of Chinese sesame oils, with mild success based on lignin (sesamin and sesamolin) content [72].

Besides adulteration detection, NIR spectroscopy has been employed for quality control of edible oils. For instance, NIR spectroscopy has been used for the determination of acidity (free fatty acid content), peroxide value (based on extinction coefficients for hydroperoxides at 232 and 270 nm which negatively correlate to NIR vibrational bands at 1440 nm), aldehyde level (via *p*-anisidine value), and level of refinement (via pyropheophytin A and isomeric diacylglycerol ratio). In particular, these parameters aid in the discernment of oil treatment, temperature of storage, and age [73], and can assist in quality assessment of waste cooking oils [74]. The latter application is of great importance to the budding biodiesel industry, where triglyceride degradation during routine cooking in open air can result in highly acidic cooking oils, requiring expensive esterification prior to use.

### 4.2. Dairy Products

Advances and practical considerations for NIR spectroscopy as a tool for the real-time (on-line or in-line) monitoring of products in the dairy industry have been reviewed recently for progress made over the last decade [75]. Likewise, a performance comparison between ultraviolet, visible, and NIR spectroscopy for off-line compositional analysis of the principal components in raw cow’s milk (fat, protein, lactose) and the rapid, online implementation thereof has been reviewed [76]. The fatty acid profile of a cheese influences both the sensory and nutritional characteristics and can serve as a diagnostic tool to distinguish cheeses of the basis of the milk employed for its production. González-Martín et al. [77] have recently demonstrated NIR as a rapid, non-destructive tool for the determination of 19 different fatty acids (from C8:0 to C20:0, including saturated and unsaturated fatty acids) in cheeses of variable composition and origins (i.e., cow, ewe, goat) by direct recording on a slice of cheese. Importantly, the authors validate the NIR methodology for reliably predicting the lipid profile of cheese, making it viable for application to samples of unknown origin.

Several important advances have been made in the detection of dairy product adulteration. Chemometric prediction of water adulteration in cow milk was demonstrated by Kamboj et al. [78] pairing NIR spectroscopy and a principle component analysis model. This model accurately predicts the adulteration of up to 2 mL of water in 5 mL of milk, providing a quick and simple determination without requiring trained analysts. A similar chemometric system was applied with NIR and mid-IR spectroscopy for the quantification of milk fat or oil (margarine, sunflower oil, corn oil, and hydrogenated vegetable oil) adulteration in yogurts [79]. Additionally, melamine adulteration in milk powder, a common target for falsification of amino acid content, was recently demonstrated as a NIR forensic technique. Therein, Mazivila et al. [80] devised a chemometric model for the NIR estimation of both melamine and sucrose in milk powder in the concentration range of 0.8–2% and 1–3% *w*/*w*, respectively. Indeed, even cow milk can act as adulterant, as is the case with the more expensive, but also more nutritious, buffalo milk [81]. In this case, a partial least squares algorithm was designed to detect cow milk in buffalo milk at wt% ratios from 10:90 to 90:10. Further, fat-filled milk powders have recently been examined for degradation using a NIR method [82]. These products, typically used as high-fat, low-cost supplements in developing world markets, are exposed to high temperature and humidity during long distribution and storage periods. A NIR method was demonstrated for the quantification of fatty acids in fat-filled milk powders derived from coconut, pal, soybean, and sunflower oils, allowing for the detection of constituent damage during accelerated storage conditions.

Finally, as a striking example of improvements made in real-time management, Muñiz et al. [83] have developed a handheld NIR reflectance spectrophotometer and artificial intelligence based mobile application which enables real-time estimation of quality parameters (e.g., lactose, protein, fat, non-fat solids) for cow’s milk. Notably, the portable NIR sensor allows milk quality parameters to be estimated onsite and employs machine-learning algorithms based on a neural network model, which is fed by NIR spectral data, offering decision-making at the farm level toward the prompt optimization of milk quality during its production.

### 4.3. Grains and Flours

NIR spectroscopy has proven beneficial for the analysis of various cereals, grains, flours, and baked goods, including specific quality parameters, which influence classification, safety, grading, and price. For example, chemometrics and machine learning have been coupled with NIR spectroscopy for the prediction of wheat quality factors [84] and the quantitative determination of fatty acid values during wheat flour storage [85]. NIR spectroscopy and partial least squares algorithms have similarly been used to determine the polyphenol content in oat grain [86]. In particular, NIR spectroscopy has emerged as an important tool to determine fraud, adulteration, contamination and provenance in grains and flours. For instance, significant instrumental improvements (e.g., hyperspectral imaging, FT-NIR) and advances in data analysis (e.g., deep learning) have allowed for the development of screening methods for detecting the presence of pests (e.g., rice weevil) across a range of stored grains, sometimes down to the individual-grain level [87,88,89].

Food fraud remains a significant problem for food regulators, importers, merchants, law enforcement personnel, and the consumer. NIR spectral data analyzed using partial least squares discriminant analysis and implementing a support vector machine algorithm, was recently shown to be a feasible method for the rapid identification of fraudulent rice varieties (5% detection limit) blended with authentic Wuchang rice samples [90]. The authors further noted that more uniform particle sizes aided in quantitative analysis (i.e., 100 mesh > 70 mesh > 40 mesh > full granules). Another subject of frequent food fraud, Gragnano pasta from the homonym Italian town, was identified against imposter products by coupling NIR spectra with two classifiers, partial least squares discriminant analysis and soft independent modeling of class analogies [91]. With a test set of 200 samples, the resulting models correctly classified all Gregnano pasta with only a single misclassification of an imposter sample. Meanwhile, the corresponding soft independent modeling of class analogies model boasts an impressive sensitivity of 6.57% with 100% specificity. NIR spectral data and partial least squares modeling identified the protein content, moisture and the types of various grains including wheat, barley, lentils, and peas, providing an expedient and non-destructive method for accurate prediction of grain quality [92]. Detection, identification, and quantification of toxins from harmful additives or inappropriate post-harvest storage or treatment can also be readily achieved by NIR methods. A recent article provides a method for detecting aflatoxins, a product of genus Aspergillus mold infestation when grains are stored under warm and humid conditions, in brown rice [93]. Detection of allergens is necessary to protect the health of consumers, and another recent paper demonstrates a matched subspace detector algorithm, coupled with NIR spectral data, to identify global adulteration of peanut in wheat flour at 0.2% [94]. Further, a recent NIR hyperspectral imaging study demonstrates the prediction of peanut adulteration in spring and winter wheat flower at a level ranging from 0.01–10 *w*/*w*% using partial least squares regression [95].

### 4.4. Chocolate and Syrups

NIR analysis has made significant recent contributions in the area of confection and sweetener analysis, moving closer to the routine quality monitoring of cacao powder cross-contamination and the authentication of honey and determination of its geographical origin, as examples. Indeed, adulteration of cocoa powder with cocoa husk, either by poor husking processes or by the intentional addition of waste processed material, is a prominent economic and health concern. A recent contribution addressed this issue by pairing NIR analysis and partial least squares discriminant analysis, accurately grouping samples of cocoa powder into sample sets of those containing less than 5% cocoa shell (the acceptance limit according to the Codex Alimentarius) and those containing between 5% and 40% [96]. Additionally, peanut flour adulteration of cocoa powder is an alarming health concern for consumers with peanut allergies. A study involving training model at proportions of 0%, 0.1%, 1%, 10% and 100% for the determination of peanut flour in cocoa powder was demonstrated recently by pairing NIR spectral data with principal component analysis and multivariate curve resolution-alternating least squares chemometrics [97]. The authors propose the use of their method for the identification and quantification of adulterants in other powders in the future.

Honey is a pure product and international regulations prohibit the addition or removal of any substance. Unfortunately, honey adulteration by unscrupulous addition of inexpensive syrups (e.g., high fructose corn syrup, beet syrup, rice syrup) is a common practice, making necessary the development of reliable analytical methods to guarantee its authenticity. For example, multi-floral honey was the target medium of a recent study for determining adulteration of honey by a variety of sugar syrups (inverted sugar, rice syrup, brown cane sugar, and fructose syrup) at ratios ranging from 5–50% [98]. The authors combine visible and NIR spectroscopy with multiple chemometrics models (hierarchical cluster analysis, linear discriminant analysis, and partial least squares regression) to detect, identify, and quantify these adulterants. In a separate study seeking to discriminate botanical origins of honey samples, NIR data was combined with partial least squares chemometrics alongside multivariate and machine learning analyses to successfully identify honey samples from multi-floral, acacia, and chestnut sources, but was unsuccessful in discriminating samples from linden sources [99]. The complex makeup of honey makes this an important and daunting challenge, as many intrinsic variables (e.g., bee species, geographical origin, production methods, storage time and temperature) complicate discriminatory identification.

### 4.5. Herbs and Spices

Food fraud in herbs and spices remains an important topic, and NIR spectroscopy, alongside Fourier-transform infrared, Raman, and nuclear magnetic resonance spectroscopies, remains an indispensable tool in the authentication of spices and the identification of adulterants, either as foreign matter (e.g., exogenous starch in paprika, curry, turmeric, or ginger powders), inferior production-related materials (e.g., non-spice vegetable matter like stamens and safflower in pure saffron), or colorants used to mask quality (e.g., synthetic Sudan azo-dyes in paprika or chili powder) [100]. Recently, adulteration of black pepper through bulking by papaya seeds, chili, and non-functional black pepper products was screened via a rapid NIR and FT-IR model, demonstrating the feasibility of combining these techniques with chemometrics to combat counterfeiting of black pepper [101]. Paprika powder, another attractive target for adulteration, was the subject of another recent study which successfully screened for potato starch, acacia gum, and annatto adulterants using a portable NIR spectrometer paired with partial least squares chemometrics [102]. Similarly, partial least square discriminant analysis paired with data acquired through diffuse reflectance NIR spectroscopy provides an avenue for detecting *Cinnamon cassia* adulteration in true cinnamon, *Cinnamon verum*, via the former’s significantly higher coumarin content [103]. Finally, adulteration of saffron, the world’s most expensive medicinal plant and high-grade spice, with lotus stamens and corn stigmas was screened by NIR spectroscopy coupled with a partial least squares discriminant analysis model [104]. The authors compared synergistic interval partial least squares, competitive adaptive reweighted sampling, and Monte Carlo uninformative variable elimination variable selection methods, and demonstrated a chemometric model for detecting adulteration of saffron in a non-destructive manner.

### 4.6. Food Additives

NIR spectral analysis has the potential to provide useful information about approved additives in foodstuffs, such as flavor enhancers, antioxidants (e.g., tertiary butyl hydroquinone E319, butylated hydroxyanisole E320, butylated hydroxytoluene E321), and food hydrocolloids (e.g., guar gum, locust bean gum, gum Arabic, carrageenans) which are key functional agents for emulsification, thickening, stabilization, and texturizing within food engineering. Quantification of ascorbic acid in soft drinks and juices has been demonstrated using NIR spectroscopy coupled with partial least squares analysis (rather than the typical methods of titration or chromatography) providing to a non-destructive and in-line option for controlling the addition of this antioxidant during the manufacturing process [105]. A similar method applied NIR and mid-IR spectroscopy for the quantification of citric acid, ascorbic acid, and the total and reducing sugar in Valência oranges, demonstrating the use of NIR spectroscopy for rapidly determining the quality of orange fruit and juice [106].

Regarding grains, first steps were recently taken toward quality assessment of gluten-free minor crop grains for total antioxidant capacity [107]. Therein, NIR spectral data was paired with Folin-Ciocalteu assay and partial least squares chemometrics to detect antioxidants in millet, buckwheat, and oat cultivars, achieving a limit of quantification of 2.6 mg gallic acid equivalents per gram of cultivar for intact seeds of the latter two grains. NIR spectroscopy was additionally applied to determine thermal degradation of soybean oil when heating in the presence of natural seed extracts (poppy seed, dehydrated goji berry, pumpkin seed, and provençal herbs) as antioxidant additives, concluding that phenolic compounds within the seed extracts act as sacrificial degradants to enhance the frying stability of soybean oil [108]. Finally, the identification and quantification of hydrocolloids, compounds that alter the properties of food when dispersed in water, is a notoriously difficult task due to the similar chemical makeup of these compounds. Recently, this challenge was overcome by pairing NIR spectroscopy with partial least squares regression and continuous locality preserving projections dimensionality reduction technique as a chemometric framework, allowing for 100% discrimination between pure solutions and blends of kappa and iota carregeenans and tara, guar, and locust bean gum [109].

## 5. NIR Spectroscopy in Agricultural Produce Analysis

Access to high quality produce is essential to human health. Thus, the accurate collection of agricultural food quality data in real-time is of utmost importance. By analyzing various components and properties of crops at various stages in their development, crop productivity can be determined early on. To maximize efficiency and lessen waste of produce, it is important that these data collection methods be non-invasive, non-destructive, and economical. Some quantitative instrumental techniques used for quality assessment of foods include gas chromatography (GC), high-pressure liquid chromatography (HPLC), or mass spectrometry (MS). However, these methods are not applicable for real-time measurements. Thus, agricultural industries have recently utilized spectroscopic instrumentation for quality analysis. One of the most common methods is near infrared (NIR) spectroscopy. NIR spectroscopy is a quantitative method that is cost efficient, prompt (real time), with simplistic technique and thus has been widely applied for food and agriculture analysis in recent years. NIR spectroscopy allows food analysts to examine the quality, composition, and the authenticity of agricultural and food products quickly and accurately. These industries rely on NIR to monitor physicochemical properties of crops throughout development, into harvest, and onto the shelves of grocery stores.

NIR is based on the electromagnetic radiation in the near infrared region (700–2500 nm). NIR spectroscopic techniques rely on the interaction of NIR radiation with matter. The chemical bonds that absorb NIR radiation are present in food and crop components such as fats, water, and carbohydrates, which can easily be detected using NIR spectroscopy. Some of the major molecular groups of interest seen in an NIR spectrum include N—H, C—H, and O—H bonds. Depending on the complexity of the sample, it is common for these bands to overlap one another. Accordingly, it is extremely essential so select an appropriate data model and instrumental mode for each application. Two major modes of NIR spectroscopy, transmission, and reflection are used in analysis depending on the physical state of the sample being tested. Transmission modes are useful when working with liquids, thin solids, and thick solids when inspecting a food item for its ripeness, or whether it contains pests or defects. Reflectance techniques are useful for measuring content in wholegrains such as protein, moisture, and oil content.

### 5.1. Fruits

Food products like fruits are often measured in reflection mode to attain information that reflect maturity and ripeness characteristics. When the NIR absorption signal is being recorded, unwanted scattering also occurs at the fruit surface which can interfere with data collection. There are several data processing techniques that are used to separate this scattering signal from the desired signal. This pre-processing correction is commonly used in the food quality assessment industry, and there are many different methods such as standard normal variate (SNV) or variable sorting for normalization (VSN). However, choosing the most optimal pre-processing method for a certain application is a challenge. Recently, Mishra et al. [110] investigated the sequential pre-processing through orthogonalization (SPORT) approach to combine multiple pre-processing methods. They applied the SPORT approach on four different fruits (apple, grape, olive, and apricot), combining multiplicative scatter correction (MSC), variable sorting for normalization (VSN) and standard normal variate (SNV). They saw improved model accuracy for all samples tested, particularly for those analyzed in reflection mode.

When analyzing fruits for quality, it can be a challenge to collect accurate quantitative data about their chemical composition. One important parameter when analyzing fruit’s quality is the soluble solids content (SSC), which is a blanket term for the water-soluble compounds in fruits. SSC is a key indicator of the internal quality of fruits and NIR spectroscopy coupled with chemometrics is a common approach to determining SSC of fruits. There are several challenges to collecting SSC data, one being the practical limitations in factory settings. This is due to the non-homogeneity of fruits in terms of size and composition as well as the difficulty in detecting single compounds from spectra. Modeling the spectra accurately requires a large amount of data that can slow down the process and limit practical use of the instrument. Guo et al. [111] developed a theoretical basis for the industrial application of NIR spectroscopy for the determination of apple SSC by using a variable selection method. They showed that competitive adaptive reweighted sampling (CARS) can simplify modeling and that competitive adaptive reweighted sampling-partial least squares (CARS-PLS) modeling can have practical applications in an industry setting. CARS-PLS model gave a correlation coefficient greater than 0.9 while other models tested showed lower value than 0.9 for correlation coefficient. Xia et al. [112] introduced a diameter correction method that allowed a more accurate and robust SSC analysis of Fuji apples. This was done by applying a calibration model based on varying fruit diameter and orientation (stem axis horizontal or vertical). They were able to lessen the effect of fruit diameter difference on the NIR spectra. One other challenge of SSC analysis on-line is the wavelength variable selection. Some commonly used methods are CARS and PLS, discussed previously. Recently, particle swarm optimization (PSO) has been applied to select optimal wavelength bands for analyzing compound percentage in fruits and is shown to outperform the more common methods. However, the PSO algorithm can suffer from premature convergence that causes significant variability in characteristic wavelength determination. Song et al. [113] implemented a piecewise particle swarm optimization (PPSO) method to extract data from NIR spectra of navel oranges for enhanced accuracy in their analysis of SSC in the fruit.

### 5.2. Grains (Rice, Cereal) and Potatoes

NIR spectroscopy is highly useful in analyzing shelf life and maturity of agricultural products like rice and potatoes. One challenge in industrial and real-time application is the optimization of portable spectrometers. There have been numerous applications of portable NIR instruments in recent years for specific analyses such as determining adulteration in oils and other food quality parameters. However, the data collection and modeling are still time consuming for portable spectrometers to be efficient in some applications. This can be potentially overcome by combining NIR spectroscopy with other analytical methods. Malegori et al. [114] developed a tandem approach of monitoring rice germ shelf life during storage using FT-NIR and a portable e-nose. This method of data convergence allowed deeper understanding of rice germ shelf life and could lead to enhanced NIR modeling of the shelf life of other food products. Le proposed a study that combines deep learning with NIR to provide a much faster method of cereal analysis than traditional NIR models [115]. The deep learning algorithm eliminates interference of NIR signal making modeling much more efficient. Jiang et al. [116] developed a portable NIR spectrometer system to dynamically monitor fatty acid content of rice during storage. They utilized an optical fiber for light capture, allowing the portable device to be used far away from the sampling site, and this makes it suitable for even harsh industrial conditions.

Another challenge in NIR spectroscopy is determining authenticity and the geographical location of certain agricultural products like grains. The major issues in mapping the genes for grain quality traits required intensive labor, time, and high cost to investigate the diversity of physicochemical traits that effect rice quality. Sampaio et al. [117] developed a robust and accurate classification model based on machine learning methods. They determined the NIR range where two genotypes of rice display major differences which allows high accuracy sorting of rice based on these characteristics. Barnaby et al. [118] correlated the grain chalk of rice to the genomic regions of NIR spectra. These spectral regions can be applied in the automation of grain chalk quantification and potentially for other grain products as well.

### 5.3. Cassava and Wheat

One of the major advantages of using NIR spectroscopy is its wide applicability. Cassava root starch content is traditionally determined by a grower snapping the root, which is destructive and not highly accurate. Bantadjan et al. [119] developed a portable NIR spectrometer in order to determine the starch content of intact cassava roots with calibration correlation (r_p_) of 0.825. They also determined that the position of spectral measurement does not affect the predicted starch content. The same group in another study investigated the application of short wave NIR spectroscopy (720–1150 nm) of both intact cassava root as well as cassava flesh [120]. They achieved a confidence interval of 95% for the two models. Carmo et al. reported the phenotyping of waxy starch cassava using Fourier Transform Infrared (FTIR) analysis on the leaves rather than the root [121]. This method has the potential to make analysis much easier for breeders, as the leaves are formed before the root.

Although NIR spectroscopy alone is a powerful analytical tool for quality control of agricultural products like cassava and wheat, recent studies have investigated how NIR can be used in tandem with other qualitative/quantitative techniques to enhance the sensitivity and selectivity of data. One study by Firmani et al. [122] utilized a multi-block strategy to determine the geographic origin of Italian semolina. The combination of sequential and orthogonalized-partial least squares-linear discriminant analysis (SO-PLS-LDA) and W-indices with NIR spectroscopy gave 100% correct classification of 140 test samples. Another study by Fan et al. [123] utilized a machine learning classification model in tandem with NIR spectroscopy to determine individual wheat seed vigor. This methodology yielded an overall accuracy of 84%, which makes the model potentially suitable for on-line application. Girolamo et al. [124] used for the first time a European (EU) validation model for screening of mycotoxins coupled with multivariate Fourier transform near infrared (FT-NIR) spectroscopy for wheat adulteration analysis. The statistical modeling used in this study enhanced the reliability of the model validation compared to screening methods based on binary results.

NIR spectroscopy has become the preferred method to analyze agricultural products and crops due to its non-destructive sampling and ease of industrial application. However, there are still some aspects that can be improved such as data modeling and timely on-line application. Researchers have recently introduced new methodologies to overcome these problems such as enhanced pre-processing strategies that can improve model accuracy. Also, combining NIR with other analytical methods (e-nose, deep learning) has been shown to enhance data collection time and simplify the modeling process overall. The developments in NIR spectroscopy for food quality in only the last year show great progress. It is expected that data acquisition and modeling methods will continue to become more accurate and less costly in the coming years.

## 6. NIR Spectroscopy in Food Supplements, Beverages, and Drinks

In modern society, people consume many different kinds of processed products such as dietary supplements, non-alcoholic beverages (fruit juices, sodas, etc.), and alcoholic beverages (wines, beers, spirits). The high demand of these products requires that strict guidelines be in place to ensure the highest quality. Manufacturers of these items utilize various analytical methods to assess purity, origin, and adulteration. Some of these techniques include gas chromatography (GC), mass spectrometry (MS), and nuclear magnetic resonance (NMR). These methods are highly accurate. However, they are costly, time consuming, and require significant sample preparation. NIR spectroscopy has recently been used to determine quality parameters of these products because it offers several advantages as it is a rapid and non-destructive technique. NIR spectroscopy is based on absorption of electromagnetic radiation in the range of 800–2500 nm. In this range, different chemical bonding environments can be identified and correlated to components in product. Some common bonds seen in NIR spectroscopy are around 1350, 2000, and 1600 cm^−1^ corresponding to C—H, O—H, and N—H bonds, respectively. The stretching and bending vibrations of these bonds can easily be analyzed by NIR spectroscopy, often in real real-time and on-line. However, the overlap of these absorbance bands can make the interpretation of complex samples more difficult. Thus, most NIR spectroscopy methods utilize chemometrics or other mathematical conversion models to extrapolate usable data from the instrument.

### 6.1. Food Supplements

The accurate detection of adulterations in food supplements is highly important as these products are often used to control ailments and dietary regulation. Thus, the introduction of additives can be potentially detrimental to the consumers. For whey protein powder samples, adulterants can currently be detected on average down to about 1–5% composition. Zaukuu et al. [125] developed a method for handheld NIR spectroscopy devices that can detect adulterant concentrations down to 0.5% (*w*/*w*) with root mean square error of prediction (RMSEP) values as low as 0.21 g/100 g. They achieved exceptional RMSEP values for samples in both commercial plastic bags and glass containers. Chen et al. [126] utilized a one-class model strategy to achieve untargeted identification of Chinese Sanqi powder adulteration. This method of class-modeling can be built for the target class without in-depth information on other classes or samples which is advantageous for authentication analysis. Thus, this method can potentially be applied for many kinds of products.

Omega-3 dietary supplements have become a highly popular food supplement product and thus, the quality screening of these is highly important. In the last year, studies have focused on applying portable spectrometers to the quality screening of these products to bring associated costs down. Karunathilaka [127] used a chemometric model based on PLS to classify the omega-3 oils based on triacylglycerol and ethyl ester polyunsaturated fats. They found that attenuated total reflectance-Fourier transform infrared (ATR-FTIR) was more sensitive than FTIR alone and that the band in the 1750–1730 cm^−1^ region (ester carbonyl group) was the most important for discriminating test samples based on their lipidic matrix. In an attempt to drive costs down further, Hespamhol et al. [128] developed a low-cost handheld spectrometer (~1000 USD) that showed potential for the delivery of in situ quality information of omega-3 supplements. This allows the raw materials to be easily quality controlled to remove adulterated precursors early in the processing stage.

### 6.2. Beverages (Fruit Juice, Soda, Energy Drink)

Commonly consumed beverages (fruit juices, sodas, energy drinks, etc.) contain many components such as sugars, nitrogen, and caffeine. It is important to utilize quantitative analysis methods that can accurately measure these components for specific drinks as well as be applied to other beverages. Petrovic et al. [129] utilized NIR spectroscopy quantified nitrogen content of grape juices with extremely high variability (over 900 juice samples). They were also able to use an independent validation set which has prevented the widespread use of this technology in similar fields. Jiang et al. [130] developed a method using miniature NIR spectroscopy scanners to accurately determine sucrose content in a multitude of everyday beverages. They achieved >99% accuracy in identifying 18 different drinks and the prototype they designed shows promise to analyze other food quality parameters.

The determination of common quality control parameters, such as soluble solids content (SSC) in the beverage industry is of high importance. Food science researchers are continuously developing new methods to enhance the speed, efficiency, and cost-effectiveness of data acquisition and analysis. Włodarska et al. [131] recently investigated the application of NIR spectroscopy in determining SSC of strawberry juice as well as the precursor fruits. The models used had high predictive ability, with root mean square error of prediction values <0.5%. Ren et al. [132] employed a method using variable combination population analysis (VCPA) coupled with NIR spectroscopy to assess quality attributes in tea. The robust hybrid VCPA algorithm allows for the application of a smartphone-based micro-Vis-NIR sensor to acquire spectral data inexpensively and on-line.

### 6.3. Alcoholic Drinks (Wines, Beers, Spirit)

Alcoholic drinks are a highly consumed product and contribute greatly to local economies. Oftentimes, beverages like spirits and wine from a certain geographical origin are adulterated with lower quality drinks. By using various portions of NIR spectrum, Hu et al. [133] were able to correlate the chemical fingerprints of Cabernet Sauvignon to their geographical location. The developed database models achieved 100% recognition rate for 374 wines from three different countries (Australia, Chile, and China). Zaukuu and coworkers combined NIR spectroscopy and e-tongue technology for the determination of adulterants in high quality Tokaj wines [134]. Partial least square regression modeling gave a coefficient of determination (R^2^) value of 0.98 and 0.97 for the e-tongue and NIRS, respectively. These methods used in tandem have practical and simple application in industry.

The storage and analysis conditions of alcoholic drinks can drastically affect quality control data obtained for these samples. Joshi et al. investigated the effect of analysis temperature for different whiskeys at a range from 25–55 °C. They saw that past 40 °C, there is a distinct alteration of absorption bands. By coupling NIR spectroscopy with two-dimensional correlation spectroscopy, they proved that this technique can dramatically improve quality evaluation efficiency over NIR alone [135]. Another group, Anjos et al. [134], found that NIR spectroscopy can discriminate aging practices of wine spirits (barrel wood species in which it was aged, aging time, etc.) with up to 90% accuracy. They found that the spectral region between 4200 cm^−1^ and 5200 cm^−1^ was most representative of sample differentiation [134].

NIR spectroscopy has been widely used for quality investigation of processed food products like supplements and beverages (alcoholic and non-alcoholic). Recently, new modeling techniques have been introduced successfully to improve these methods. In terms of adulterant detection, models have been developed that are more accurate and more applicable to a wider range of products. Some new models even show potential application in smartphones. In a similar regard, portable spectrometers have also been applied to more diverse areas with competitive performance relative to benchtop instruments. These advancements are essential in lowering the costs and time associated with quality control analysis.

## 7. NIR in Food and Pharmaceutical Raw Materials

In the realm of food and pharmaceuticals, NIR spectroscopy has reached a point to where it is readily available in both screening technologies and in process monitoring applications [136,137]. As a screening technology, NIR can be used in a laboratory setting and at “point-of-use” as a portable technology [138]. Laboratory applications are typically performed using a benchtop instrument, which are commonly available using the Fourier-transform approach. Some instruments may also be dispersive, but usually these types of spectrometers are portable/handheld/small footprint [139] and used in most point-of-use technologies [140].

NIR when applied to screening applications is used to make rapid determinations, usually between a few seconds to one minute, and the methods are often combined with either a library-type comparison or an analysis made using a multivariate model [141]. Regardless of the algorithm, the spectrum obtained in the field is compared to a known library or model that contains reference samples. Until recently, this comparison has been made mostly on the local computer that is available on the portable NIR spectrometer but recent advances in server architecture have made it so that possible for certain applications to compare spectra under study in real-time to a virtual server via a wireless network connection or cloud. This new area of innovation means that models and libraries can constantly be adapted to the needs of users and undergo refinement in real-time such as monitoring of medication taken by elderly patients [141].

The use of NIR in process controls [142] is increasingly common and is has been reported in applications such as the manufacturing of botanical drug products. This class of products has intrinsic variability in the raw materials and ingredients may undergo complex chemical reactions during production. This makes it imperative to perform real-time monitoring of the reactions, such as in the case of production of compound danshen dripping pill (CDDP) reported by Zhang et al. [142], where an in-line NIR method was developed for commercial production of the botanical product. In almost all in-line/at-line/on-line applications the NIR method requires careful development and validation. This includes the creation of a representative calibration set and validation set. The models are usually chosen such that an appropriate number of batches, containing representative variability expected in the process, are used in the model development. This is a key consideration for the success of any NIR model. For example, in a recent study by Caroço et al. [140] aimed at assessing raw citrus-related materials for pectin production, the research team built a design space of 85 raw material samples that contained representative fruits, including 43 lemon, 27 lime, and 15 orange peel samples. The ultimate design space depends on the application, but especially for naturally-derived raw ingredients, there are known sources of variability [140] that should be built into the model robustness.

The use of spectral ranges [143] and pre-processing [144] also help to enhance model performance [145] and the method parameters such as those commonly reported for chromatographic methods [146] are also reported: specificity, accuracy, linearity and precision. For process NIR applications, some of the areas where NIR brings considerable utility is in moisture content, particle sizing, form identification, density, and blend uniformity [137]. NIR has also been useful to monitor coating thickness [147], and in a recent application was used as part of a combined-data-approach to predict the active ingredient dissolution performance of enterically-coated microspheres [148]. Finally, there are also new approaches to the critical problem of calibration transfer, which is the use of a model or method that was developed on a different—often central—geographically distributed transfer instrument. This is important since all instruments contain small differences and an effective calibration transfer [149] approach obviates the need to develop a unique model on each transfer instrument. Skotare et al. [150] recently reported on the use of a novel approach called joint and unique multiblock analysis (JUMBA) to achieve instrument standardization to transfer models effectively compared to traditional approaches.

## 8. NIR Spectroscopy in Meat and Meat Products, Fish, and Seafood Products

Muscle foods, including meat and fish, are very important from the point of view of human nutrition and commercial activity worldwide [151]. There have been significant efforts, both from the industry and academia, on improving quality and quantity of the raw and processed muscle food types in recent years [151,152,153,154,155]. Accordingly, there have been increasing demands for the development robust and time effective monitoring techniques to track compositional and safety compliance issues as conventional chemical methods of meat quality assessment become more expensive and time consuming. Near infrared spectroscopy (NIR) has drawn considerable interest for many reasons, including its portability, speed, cost-effectiveness, suitability for in-line as well as on-line analysis, and the ability to concurrently measure different parameters for a large number of samples [151,152,153,154,155,156,157]. Several studies reported on the use of NIR for meat quality assessment and the analyses of meat properties in live animals or carcasses due to the capability of IR light to penetrate several layers of tissue [151,152,153,154,155,156,157,158,159]. Two published papers demonstrated that the short wavelength radiation in the NIR region of the electromagnetic spectrum can potentially be utilized to non-invasively assess meat quality [152,154]. The results of these studies are significant because of the limited literature non-invasive analysis of crucial nutrients in the live animals; thus, these reports fill that gap.

Recently, Simon et al. [152] demonstrated NIR spectroscopy as a non-invasive technique to assess the nutritional parameters of spiny lobsters. The authors also used the Bruker MPA FTIR spectrometer to determine abdominal muscle composition and hepatopancreas composition. Total carbon content (AM_C_) and total lipid content (HP_TL_) models developed by the authors in this study demonstrated that NIR has great potential for rapid non-invasive screening of live lobsters and other fish as well as poultry products. For fisheries and poultry farms, this technique will allow for monitoring the condition of live stocks during changes in feed availability, feed quality, water temperature as well as in the wild and in cultured conditions. The NIR HP_TL_ model would be effective in determining the growth in the number of livestock with no requirement to capture or tag individual animals. The NIR AM_C_ analysis that resulted in low abdominal carbon content is likely associated with low meat yield, lower health benefits from decline in the omega-3 fatty acid levels, and a likely reduction in taste and texture quality. Additionally, the authors have been able to establish a good correlation between NIR AM_C_ as well as haemolymph total protein (TP) models, demonstrating NIR’s ability to predict livestock vitality crucial for live shipment.

Caballero et al. [160] have recently demonstrated a fast and accurate method to determine crucial nutritional parameters, including protein, lipid, slat, and carbohydrate contents, based coupled NIR and data mining techniques in an automated manner. The authors have conducted a study on two batches of Iberian pork meat products. The authors divided one batch of each product into training as well as validation sets. The authors have accessed the prediction equations from the NIR, while data mining was implemented to obtain the nutritional data for the training batches. Finally, NIR data from the validation batches were used to evaluate the prediction equations. The authors have demonstrated that the NIR based evaluation method that they have developed is many times faster than the conventional approach taking only 10 min as opposed to the orthodox six-day process.

Pochanagone et al. [157] have also demonstrated that a NIR-based method can be used to rapidly determine the salt content in tuna fish without having to go though the time consuming conventional wet chemical assay. The salt content, predicted by the NIR model, was observed to be very close with the one obtained from the conventional assay providing a 96% confidence interval. Although, the salt is IR inactive, the authors have explored the influence of salt on the absorbance of the NIR energy providing access to a way of determining the salt content by exploiting the changes in the water band at 970 nm. Calibration equations for these experiments were derived from the frozen fish pieces as well as ground samples by employing linear regression analysis for the wavelength region of 700–100 nm. The same authors have also demonstrated the ability of the NIR based method to rapidly determine in tuna, the content of histamine, a neurotransmitter responsible for the inflammatory response in humans [153]. In this study, the authors have coupled the dry extension system for IR (DESIR) with the NIR to determine the histamine in tuna at the ppm level. Their findings showed that the DESIR method significantly improved NIR application allowing the detection in the ppm level. This method is still not fully developed, and authors are working on it.

Kilbo et al. [158] have demonstrated the ability of NIR based methods to rapidly and precisely determine biogenic amine content as the same could be used to evaluate the freshness of fish and poultry products. The authors, however, have pointed out that dehydrated samples must be prepared, following procedures that they have developed, as the developed NIR method is not suitable for the samples with high water content. There are a few other recent reports demonstrating the use of a NIR based analysis method for rapid and cost effective identification of crucial nutrients with high precision rates [161]. For example, recently, Miller et al. [159] reported a NIR based method for quick assessments of various components of king salmon and greenshell mussels. Most of the demonstrated methods take minutes to prepare and analyze the samples as compared to the days required for the conventional wet-lab chemical assays. Therefore, NIR techniques seem to be highly promising for the fish and poultry industries and should be extensively looked into to develop fully from the perspective of time as well as cost effectiveness.

## 9. Chemometrics Approach and Multivariate Analyses of Spectral Data Analysis

Beer’s law is widely used in analytical spectroscopy to correlate the concentrations of standard solutions with corresponding analyte absorbances to construct the calibration curve that is later utilized to determine the concentration of analyte of unknown samples. Beer’s law relies on the use of standard solution absorbances at one wavelength (typically at lambda (λmax), a process known as univariate spectral analysis. Nonetheless, Beer’s law and univariate spectral analysis are ineffective for reliable and accurate sample analysis where there is a considerable blue or red spectral shift at lambda max. Variation in other wavelengths/wavenumber regions is often not considered but contains significant data that may be utilized to map analyte absorption fingerprint signatures and spectral profiles for ultimate pattern recognition and/or quantification of analytes in unknown samples. Moreover, univariate spectral analysis is incapable of multicomponent analyses of multiple analytes without tedious spectral resolution or deconvolution.

Multivariate regression methods of the analysis of spectral data were developed to address the challenges and shortcomings of univariate spectral analysis. In general, multivariate regression analysis of spectral data involves applying statistics and advanced mathematics for processing spectral data for chemical analysis. Multivariate regression analysis allows for the simultaneous investigation of multiple wavelengths/wavenumbers or sections of wavelengths/wavenumbers for chemical analysis. Moreover, multivariate analyses are capable of multicomponent analyte sample analysis without the requirement for spectral resolution and spectral deconvolutions.

The scope multivariate regression analyses is broad and can be classified into two major categories [162,163,164,165,166]. The first category of multivariate regression analyses focuses primarily on the elucidation of the structural relationships in a data set that may facilitate pattern recognition and sample classifications [162,163,164,165,166]. Principal component analysis (PCA) is most commonly used for sample pattern recognition. PCA relies on the decomposition of x-variables (spectra data in spectroscopy) and represents the data set in a new orthogonal coordinate system, eliminating the collinearity between the x-variables. The first initial few principal components (*PCs*) typically contained the maximum variability and the most useful information in the data set. Higher *PCs* often contained insignificant information or “noise”. The use of few *PCs* to represent the data is advantageous since it reduces data dimensions.

Samples are grouped on PCA score plots based on the similarity or dissimilarity in their physical and chemical properties. For instance, samples with similar properties origin, geographical location or composition are often grouped close to one another on the PCA scores-plots [162,163,164,165,166]. PCA scores plots may reveal hidden data or information that may not be apparent from ordinary data examinations. Nonetheless, the capability and effectiveness of PCA for accurate sample classification is very limited. To address these challenges, linear discriminant analysis (LDA), Fisher linear discriminant analysis (FLD), and soft independent modeling by class analogy (SIMCA) have been developed for sample differentiation and classification with excellent accuracy [162,163,164,165,166]. Moreover, artificial neural networks (ANN), random forests (RF), support vector machines (SVM), deep convolutional neural networks (DCNN), and 3D convolution neural networks (3D-CNN) are also excellent for data processing and efficient sample classification.

The second category of multivariate analysis focuses on using regression analysis for process optimization, process or system control, sample calibrations, and instrumental calibration [162,163,164,165,166]. Partial-least-squares (PLS) regression remains the most widely used regression analysis model for the sample and instrumental calibrations. However, other regression analysis techniques, including principal component regression (PCR), variable selection random frog partial least squares (RF-PLS) algorithms, and others, have been strategically developed and used for sample and instrumental calibrations. Regardless of the regression method, the initial stage of regression model development is model optimization and refinement. The overarching goal of any multivariate regression is to predict future samples’ analyte concentration with a degree of certainty and good accuracy using a process known in multivariate analysis as “validation”. The developed regression models must be adequately validated, usually with independent validation samples of known concentrations. Root-mean-square-error-of-prediction (*RMSEP*) and root-mean-square-percent-relative error (*RMS%RE*) are often used to evaluate the reliability and performance of the regression model for accurate determination of analyte concentrations of validation or future samples.

## 10. Multivariate Analyses of NIR Spectra for Selected Food Quality Assessment

Quality assessment of the nutritional values of meat, pork, fish, and egg and the authentication of meat, pork, fish, and egg products is of considerable importance directly impacting the human diet with financial implications for meat industries. As a result, the development of rapid protocols for assessing the quality of meat continue to be of a significance interest to the food processing industry. Unlike the conventional methods of meat analysis, NIR spectroscopy is fast, low-cost, non-destructive, and readily available. Some of the recently reported multivariate analyses of NIR spectra meat, pork, fish, and egg quality assessments are discussed in this section.

### 10.1. Meat and Pork

The practical applications of NIR spectroscopy for quality checks and assessment of meat products have been published [167,168]. The protocols were capable to segregate meat tissues (lean and fat) and to predict protein, moisture, fat, and fatty acids content, and the sheer force of meat in food processing industry [167,168]. The use of a combination of NIR spectroscopy in conjunction with PLS-DA models for rapid differentiation between South African game species, irrespective of the treatment (fresh or previously frozen) or the muscle type was also demonstrated during the review period [169]. The result of this study is significant because of its capability to distinguish between fresh and previously frozen meat (accuracy >90%). Importantly, the protocol differentiated ostrich muscles, the forequarters and hindquarters of the zebra, and springbok muscles, with an accuracy of 100%, 90.3%, and 97.9%, respectively.

The adulteration of meat products is wide-spread and a challenge in the food industry. Besides the economic losses for the food industry, adulteration of meat also has a cultural and religious consequence. For example, the adulteration of beef meat with pork negatively impacts people who are forbidden from the intake of pork meat due to cultural or religious beliefs. Several studies reported on the use of multivariate regression analysis of NIR spectra for meat quality assessment during the review period. For instance, a combined use of a Vis/NIR reflectance spectrometer with a support vector machine (SVM), random forest (RF), PCA, PLS regression, and deep convolutional neural network (DCNN) for the accurate detection of adulteration of minced beef was reported [170]. The use of DCNN and PCA allowed the identification of beef adulteration type with excellent accuracy (>99%). The obtained figure-of-merit (R^2^ = 0.973) demonstrates the linearity of the PLS regression. Importantly, the PLS regression accurately predicted the adulterated beef with pork meat with a root-mean-square-error-of-prediction (RMSEP) of 2.145.

In a related study, the use of NIR spectroscopy in conjunction with PLS-DA for quantitative detection of binary and ternary adulteration of minced beef meat with pork and duck meat was reported [171]. According to the study, the use of DA models with selected wavelengths (none with preprocess methods) resulted in accurate classification rates for binary (100%) and ternary systems (91.5%). Moreover, the developed PLS regressions resulted in R^2^ > 0.95 and RMSEP as low as 7.27, demonstrating the linearity and accuracy of the protocol for the detection of adulterated minced beef meat. Application of NIR spectroscopy combined with PCA and PLS-DA for efficient discrimination and reliable detection of fraud in minced lamb and beef production was also reported [172]. Moreover, the use of NIR and deep 3D convolution neural network (3D-CNN) for red meat classification was demonstrated with a remarkable accuracy (>96.9%) [173].

The authentication, quality assessment, and assurance of pork meat also generated considerable interest during the review period. For instance, an analytical protocol involving the combined use of NR reflectance spectroscopy and PCA and PLS-DA for rapid analysis, detection, and quantification of pure pork and pork meat that were adulterated with other meat samples was published [174]. According to the report, PCA and PLS-DA notably achieved accurate classification of pure and adulterated pork meat samples. In addition, the developed PLS regression predicted the pork content of independent validation samples, with good accuracy and a low RMSEP value of 1.84%. The use of a portable FT-NIR instrument coupled to a 5-m fibre optic sensor head for the accurate prediction of moisture, protein and fat in Iberian pig pork loins [175] and the authentication of Iberian ham quality were reported [176]. Other significant reported studies on the quality assessment of pork meat included the classification and prediction of nutritional parameters for different iberian pork meat products (e.g., dry-cured ham) [177], characterization of sous vide pork loin [178], and classification of brined pork samples and the prediction of brining salt concentrations [179]. Other related published studies include the rapid detection of total volatile basic nitrogen content in frozen pork [180] and the detection of bacterial foodborne pathogens in fresh pork muscles [181].

### 10.2. Fish and Eggs

Fish continues to be a significant source of protein and nutrients in the human diet. Assessment and authentication of the food quality of fish therefore continues to generate interest during the review period. For instance, the use of NIR reflectance spectroscopy and PLS regression for fast determination of the textural properties of silver carp (Hypophthalmichthys molitrix) has been reported [182]. The accuracy of the protocol for the determination of fish flesh textural properties has been demonstrated by the reported figure-of-merit of the PLS regression and low RMSEP for water holding capacity (RMSEP, 0.10), hardness (RMSEP, 0.54), resilience (RMSEP, 0.08), springiness (RMSEP, 0.96), chewiness (RMSEP, 2.63), and shear force (RMSEP, 0.41). In a related study, the capability of NIR hyperspectral imaging for the accurate determination of total volatile basic nitrogen content and the characterization of the fish textural profile and fish freshness has been published [183]. The result of the study is significant because spectra from fish eyes and gills can potentially be used for the prediction of total volatile basic nitrogen content and the characterization of textural profile analysis of intact fish. Another study reported on the combined use of NIR and PLS regression for the determination and evaluation of texture and freshness (pH, total content of volatile basic nitrogen, thiobarbituric acid reactive substances, and ATP-related compounds) of bighead carp (Aristichthys nobilis), with excellent accuracy [184]. The combined use of NIR spectroscopy, PCA, and PLS-DA for fast authentication and classification of European sea bass (Dicentrarchus labrax L.) according to production method, farming system, and geographical origin has been reported [185]. According to the report, PLS-DA models achieved an impressive classification rate of 100% for both wild and farmed sea bass. In addition, the reported PLS-DA models correctly classified sea bass according to production method with impressive accuracy.

In addition to meat and fish, eggs are also a critical component of the human diet and a major sources of protein and minerals. Eggs are widely consumed in a variety of ways such as boiled, fried, or scrambled. Eggs are also a critical component or ingredient of other food products including breads and cakes. Quality assessment and authentication of eggs and egg products is therefore imperative for the food industry. A fast and accurate protocol involving the use of ATR-FTIR and NIR spectroscopy in combination with PCA for liquid egg authenticity and possible adulteration (with water) has been reported [185]. An accurate method based on UV-VIS-NIR spectroscopy utilizing support vector machine classification as well as linear and quadratic discriminant analysis for rapid fraud detection in hen housing systems declared on egg product labels has also been published [186]. According to the study, unlike cholesterol concentration, egg lipid extract content was found to be a very promising tool for the analytical verification, classification, and authentication of an egg farming system. A related study reported the use of NIR spectroscopy and PCA for the non-destructive identification of native eggs in Chinese markets [187]. Quality checks and assessment of eggs for cholesterol content from fresh manually shelled egg yolks or pasteurized pre-shelled egg yolks by the combined use of UV-Vis-NIR and PCR and PLS regression [188] or UV-Vis-NIR and artificial neural networks (ANN) [189] have been reported.

## 11. Multivariate Analyses of NIR Spectra for Selected Food Quality Assessment

Near infrared spectroscopy (NIR) is a rapid, non-invasive, and non-destructive technique that allows investigators to collect powerful analytical information from a sample. For these reasons, this analytical technique has been widely employed for food quality assessment at different production stages of several products, such as dairy products, juices, edible oils, fruits, seeds, and others [190,191,192,193,194]. However, this technique presents certain challenges, such as an incredible amount of molecular information, spectrum variations between samples, and overlapping spectral bands due to the presence of several components in the samples, as well as other problems [38]. To overcome these difficulties, NIR spectra may be processed using chemometric and multivariate analyses to obtain valuable information such as classification of samples and/or quantification of compounds within them. As a result, NIR spectroscopy coupled to multivariate analysis and chemometrics tools is a valuable approach for the evaluation of quality as well as quantification of nutritional compounds in food products and classification and discrimination of food samples into different groups.

### 11.1. Dairy Products

Consumption of dairy products is promoted by the medical and scientific community because it is a high source of calcium, proteins, and fatty acids [195,196]. For this reason, evaluation of the quality and authenticity of dairy products is of great importance to ensure the presence of essential nutrients and to guarantee that these products will not present a risk to consumer health. The Codex Alimentarius is a series of international guidelines and codes to maintain food standards. Moreover, these guidelines set the regulatory standards for use and concentration of various additives, such as conservatives, stabilizers, antioxidants, etc., in products. However, these regulations are usually violated by dairy companies with intent to increase profits [197,198]. As a result, different adulterated dairy products such as milk, cheese, and butter are frequently found in the market. Consequently, the in-line and off-line evaluation of these types of products are of great importance. Therefore, application of NIR spectroscopy coupled with multivariate analysis to develop new quality control methodologies for dairy products has been deemed of great importance within the scientific community.

Various groups have developed NIR spectroscopy methods for quality control of milk samples [192,199,200,201,202]. Milk, the principal consumed dairy product, could be adulterated in several ways by addition of different substances such as urea, hydrogen peroxide, vegetable proteins, water, and others [203,204]. Teixeira et al. [200] evaluated adulteration of goat milk with urea, bovine whey, water, and cow milk as adulterants. In this work, 300 authentic goat milk samples and 300 adulterated samples were analyzed using NIR spectroscopy. These adulterated samples were created by addition of four adulterants at different concentrations (1, 5, 10, 15, and 20% *V*/*V*). Before classification, NIR spectra were preprocessed to eliminate negative effects of light scattering through application of standard normative variance (SNV) and multiplicative scatter correction (MSC). NIR spectra were then centered to the mean through application of Savitzky–Golay derivative. Following this, principal component analysis (PCA) was applied to these data, and Q or T^2^ Hoteling residue values were employed to evaluate distribution of samples and to eliminate possible outliers. Researchers divided these samples, where two thirds of them were employed to build the model, while the remaining third were employed as validation samples. Three different algorithms, namely k-nearest neighbor (k-NN), partial least square-discriminant analysis (PLS-DA) and soft independent modeling of class analogies (SIMCA), were employed by the authors to achieve classification. Performance of the models were evaluated through use of sensitivity and selectivity. First, algorithms were applied to classify samples into two classes: authentic and adulterated samples. Then, algorithms were employed for discrimination of the samples into five classes: authentic, milk-adulterated, urea-adulterated, bovine whey-adulterated, and water-adulterated samples. Classification results obtained with all three models were reliable and capable of discriminating between authentic and adulterated samples. However, based on the three algorithms employed, PLS-DA provided the best performance in both types of classification (two and five classes), reaching 100% sensitivity and selectivity for cross-validation and prediction samples.

Another research group analyzed the adulteration of goat milk with cow milk, as well as the protein and fat content in the samples using NIR coupled PLS algorithms [198]. In this work, Pereira et al. collected goat and cow milk samples and randomly adulterated them generating a total of 112 samples. Before calibration, NIR spectra were preprocessed employing the following techniques: linear baseline correction (LBC), baseline offset (BO), standard normal variate transformation (SNV), and multiplicative scatter correction (MSC). Pereira et al. first applied PCA to classify samples into three groups: goat, cow, and adulterated samples. Researchers observed that the first five principal components (PC) accounted for 99% of the variance. As a result, the use of only PC1 and PC2 was not enough to discriminate the samples into the original three groups. To resolve this problem, Pereira et al. applied PLS-DA to the data, achieving 100% sensitivity, accuracy and specificity for classification of samples. Another type of classification into these three classes was performed using fat and protein content in the samples, but this information was not enough to reach accurate discrimination. To overcome this issue, these scientists analyzed NIR spectra using internal PLS (*i*PLS) and Successive Projections Algorithm for interval selection in PLS regression (iSPA-PLS) to develop regression models to quantify fat and protein content in those milk samples. As a result, these researchers achieved good classification with iSPA-PLS, employing 11 latent variables that contained spectral information corresponding to water, protein, and fat content.

NIR spectroscopy coupled with chemometrics has been employed by Strani and coworkers [205] to analyze the renneting conditions of milk to produce high quality cheese. In this study, they evaluated the effect of temperature, pH, and fat concentration on the renneting procedure through a Box-Behnken experimental design. NIR spectra obtained from 17 trials were analyzed with interval-PCA (i-PCA) and analysis of variance (ANOVA)-simultaneous component analysis (ASCA). From this study, i-PCA model allowed Strani et al. to predict how the renneting procedure will be affected with different operating conditions. A similar work published by the same group, analyzed NIR spectra and rheological data of fifteen samples at various renneting conditions with multivariate curve resolution—alternating least square (MCR-ALS) to build multivariate statistical process control (MSPC) charts [201]. The analysis performed with MCR-ALS demonstrated that three components represented the major phases of the renneting procedure under different cheese fabrication conditions. After this, these researchers built MSPC with the calibration matrix from MCR-ALS and profiles from PCA to obtain T^2^ and Q residuals values. The accuracy of these charts was evaluated through use of sensitivity and specificity values. The Q residuals values presented a better performance than T^2^ residual values with specificity and sensitivity values of 94% and 100%, respectively. These results proved that the evaluation of NIR spectra, as well as rheological properties of the renneting process, could be followed in-line. Thus, allowing manufacturers to change the production conditions in real time in order to produce a high-quality product.

Visconti et al. [190] evaluated the adulteration of grated hard cheese using NIR spectroscopy coupled to discriminant classification techniques such as k-NN and PLS-DA and class modeling techniques SIMCA. Based on all models studied, PLS-DA yielded the greatest discrimination results between authentic and adulterated cheese samples with high values of sensitivity, specificity, and precision. Bergamaschi et al. [206] analyzed the content of fatty acids (FA) in cheese that were prepared with milk sourced from cows raised in different farming systems, employing NIR spectroscopy with partial least squares regression. Results obtained by this group allowed for the quantification of FA content in validation samples with high accuracy, and the model developed was deemed to be a valuable tool for the determination of FA in unknown samples and identify the milk source [206].

### 11.2. Edible Oils

Edible oils are important products for human consumption because they provide fatty acids, energy, and other nutrients. Moreover, some vegetable oils contain certain molecules with biological properties. Some of them include anti-carcinogenic effects, reduction of cholesterol absorption, and decrease in blood levels of low-density lipoprotein (LDL) [207,208]. For these reasons, several scientific groups have focused on development of NIR spectroscopy methods for evaluation of quality, as well as determination of various components present in edible oils [194,209,210,211,212,213]. Liu and co-workers [209] developed a chemometric method using NIR spectroscopy for quantification of phytosterols in peanut, corn, soybean, and colza oils. In this work, authors evaluated 62 oil samples through NIR spectroscopy and gas chromatography coupled to mass spectroscopy (GC-MS). These samples were divided into the two groups of calibration and prediction samples. Spectra from calibration samples were employed to examine three optimized PLS calibration models obtained through different preprocessing methods. Good values of correlation coefficient (R) and ratio of prediction to deviation (RPD) indicated high accuracy for each calibration model. Moreover, concentration values obtained for prediction samples were compared to those obtained from GC-MS. Slopes of both calibration techniques employed by these authors are in good agreement, demonstrating that PLS calibration models could be employed as a non-destructive, fast, and easy methodology for the quantification of phytosterols in edible oils. Another research group investigated the free fatty acids (FFA) content in palm oil using a portable NIR spectrometer coupled to unsupervised and supervised multivariate analysis [210]. In this case, Kaufmann et al. were able to discriminate palm oils samples with different contents of FFA through linear discriminant analysis (LDA) and k-NN algorithms achieving 100% accuracy.

Another edible oil commonly used is olive oil. Several groups have investigated the quality and nutritional properties of this type of oil [194,211,213,214]. For example, Jiménez-Carvelo and coworkers evaluated discrimination of extra virgin olive oils (EVOO) from Argentina according to their registered denomination of origin (RDO) and adulteration through combination of NIR and fluorescence spectroscopy coupled with multivariate analysis [195]. Jiménez-Carvelo et al. demonstrated that first order NIR data processed using PCA followed by PLS1-DA, and second order fluorescence data processed using both PCA and a multidimensional version of PLS1–DA (NPLS–DA), allowed for the classification of the RDO of each sample with 100% accuracy.

### 11.3. Agricultural Products

Due to an increase in world population, a concomitant increase on demand of high quality agricultural products is of major concern [215]. For this reason, several research groups have been focusing their attention on developing simple, fast, non-destructive, and low-cost methodologies for evaluating quality of agricultural products. For example, Santos and co-workers employed NIR spectroscopy to determine if sorghum grains were infested with insects [216]. This model, developed using PLS-DA, was able to discriminate 100% infested from non-infested samples. In another work, Biancolillo et al. evaluated the insect infestation of stored rice using NIR spectroscopy coupled to PLS-DA and SIMCA to analyze infested and edible samples from different countries [217]. Models developed by Biancolillo and co-workers allowed for the classification of the two samples with high accuracy. Moreover, the PLS-DA model allowed independent discrimination of infested samples by country of origin.

Another concern within the global community is the availability of organic products [218,219,220]. As a result, investigations on the presence of pesticides in these agricultural products is also important. Yazici and partners developed a NIR-based prediction model for detection of pesticide residues in strawberry samples [218]. In this case, the authors built a robust PLSR model for quantification of two pesticides that was in good agreement with values obtained through LC-MS/MS. In addition, Xiao and coworkers performed a pilot study in China to discriminate between organic and conventional rice samples through NIR spectra coupled to PCA and PLS regression [220]. The model obtained after NIR spectra were preprocessed demonstrated discrimination capability between these two types of rice samples.

NIR spectroscopy has proven to be a simple, yet commanding tool that can provide an overwhelming amount of information. When coupled with chemometric tools and multivariate analysis, this information proves even more useful for providing scientists with detection and discrimination of molecular components. Overall, these strategies of pairing NIR spectroscopy with chemometrics and/or multivariate analyses have proven to be a powerful, non-destructive, rapid, and simple analytical technique that may be employed for quality control of food products in either an in-line or off-line production procedure.

## 12. Quartz Crystal Microbalance (QCM) Coupled with Multivariate Analyses for Food Quality Assessment

### 12.1. Quartz Crystal Microbalance

Quartz crystal microbalance (QCM) systems present a valuable alternative to standard analytical methodologies for quality food assessment. The QCM offers major advantages such as economic feasibility, rapid sample processing, high analyte sensitivity and selectivity, and portability, among other properties. Samples may be processed via either liquid or gas phase analyte adsorption onto a coated quartz crystal resonator (QCR) and further monitoring of changes in frequencies. Another advantage of this approach lies in the ability to reuse coated sensors as a result of the ease of desorption of analyte from the coating material. There have been several recent reviews describing electronic operation of QCM and further exploration of coating-specific VOC detections in greater detail [221,222,223]. As a result of the gravimetric mechanism of operation [224], a multitude of coatings have also been explored for analyte specificity and multi-sensor array (MSA) development [225]. In a similar manner, multiple harmonic analyses may also lead to one sensor providing analyte specific responses to complex mixtures that result in production of a virtual sensor array (VSA) [226,227,228,229,230,231]. Thus, QCM has the potential to be a uniquely useful analytical tool for food quality analyses [232]. In the following sections, we explore recent developments of liquid phase and VOC detection and discrimination in QCM-based techniques for food quality analyses.

### 12.2. Liquid-Phase QCM Advances

As noted in the previous section, many industries employ NIR spectroscopy for food quality control. However, other emerging, non-destructive methodologies are also under consideration in this area. Liquid phase QCM systems are another method that scientists have explored to investigate quality control in food samples and detection of proteins and pesticides [233,234,235,236]. Gelatin is an animal protein widely used in confectionary products [237], and has become more popular as a gelling agent in food and pharmaceutical industries [238]. In some cultures, the source of gelatin (from porcine or beef) is strictly prohibited for consumption and certification of analysis is required for consumers to assess before purchasing [237,239]. For this reason, and owing to its high sensitivity and ease of operation, Muharramah and coworkers have employed a liquid phase QCM system for determination of gelatin sources from ice cream samples [239]. These investigators electropolymerized an aniline solution in dilute hydrochloric acid onto a gold QCR using cyclic voltammetry. Two types of gelatins, from bovine and porcine sources, were employed in production of two ice cream batches. After processing both ice cream samples, polyaniline coated QCRs were exposed to dilute aqueous solutions of each batch. Investigators observed that ice cream samples prepared with bovine gelatin presented positive frequency shifts, while batch samples containing porcine gelatin resulted in negative frequencies. Thus, investigators demonstrated that electropolymerized polyaniline coating successfully detected different sources of gelatin samples [239].

Another protein contaminant that often leads to health and cultural concerns is pork serum albumin (PSA) [240]. As an incentive to reduce costs, manufacturers often mix pork products with beef or other meats [241]. In an attempt to address these potential adulteration concerns, Cheubong and coworkers developed a novel molecularly imprinted nanogel (MIP-NG) coating as sensing material for PSA [242]. After preparation of MIP-NG sensors, these investigators optimized sensor responses with the dilution factor of real beef samples and investigated the binding behaviors of PSA with MIP-NGs. Finally, these researchers conducted a controlled study of adulterated beef sample, where PSA amounts were varied, and QCM responses were collected and analyzed. They determined that the limit of detection (LOD) was 1%, or 12 μg/mL, of pork protein. Thus, these investigators demonstrated the utility of MIP-NG as a useful coating for QCRs to provide the sensitive, rapid detection of porcine protein adulteration [242].

Other investigators have explored the utility of MIPs for sensing small molecules, such as pesticides, using liquid phase QCM [243]. In 2019, Cakir et al. [243] used a 2,4-dichlorophenoxyacetic acid (2,4-D) imprinted ethylene glycol dimethacrylate-N-methacryloyl-(L)-tryptophan methyl ester polymer film (p[EGDMA]-[MATrp]). The herbicide, 2,4-D, is commonly found as a contaminant in food, soil, and ground water. When examining the selectivity of this novel MIP sensor with other structurally similar compounds, such as 2,4,6-trichlorobenzoic acid and 2,4-dichlorophenol, selectivity toward 2,4-D remained highest. The linearity range and LOD were also determined to be 50–1770 ng/L and 24.57 ng/L, respectively. When these investigators spiked real samples of apples at 250, 500, and 1000 ng/L, they determined that no false positives were obtained and there were low relative standard deviations, which demonstrated high accuracy and precision for this method [243].

In another report, Sroysee and coworkers [244] designed two new MIPs for detection of carbofuran (CBF) and profenofos (PFF). These compounds were chosen for this study because of their widespread application as pesticides and detrimental effects to the environment and to the health of humans and animals. CBF-MIPs were prepared using methacrylic acid and ethylene glycol dimethacrylate (MAA-EGDMA), while PFF-MIPs were fabricated from polyurethane-based poly(4-vinylphenol) (PUPPVP). Concentration-based studies were performed to determine linear ranges for each analyte. Analyses of CBF-MIPs have a linear response range between 0.5 and 1000 μM, and PFF-MIPs responded in a linearly in the range of 5–1000 μM. LODs for CBF and PFF were determined to be 0.21 and 0.38 μM, respectively. In this regard, these investigators successfully demonstrated the selectivity and sensitivity of their MIP QCR sensors.

### 12.3. Gas-Phase QCM Advances

Gas phase analysis of VOCs is also an area of increasing interest. Many plant-based fragrant components, such as secondary metabolites, essential oils, polyphenols, terpenes, and other compounds, are volatile compounds that may be used as early onset indicators for age, health, and methods of plant production [245,246,247,248,249]. In food products, they are often present as odor indicators for food freshness, shelf life, and/or pathogen contamination [250,251,252]. Mangos, for example, are the fifth most commercialized fruit around the world [253]. They contain high amounts of vitamin C, fiber, and are commonly free from heavy metals. Mangos are also rich in terpenes, ketones, and hydrocarbons [254], and different mangos may have slightly different flavors depending on the presence of a variety of VOCs [255]. Recently, Ghatak and coworkers developed a coating based on mustard oil for identifying an unsaturated alkene VOC component of mango fragrance, known as ocimene [256]. Other vegetable oils were studied in comparison to the sensing performance of mustard oil. However, mustard oil was determined to have the highest sensitivity and resulted in a low LOD of 1.04 ppm. These researchers hypothesized that this was principally a result of the unique mixture of saturated and unsaturated compounds present in this oil. Results obtained with this sensor were also determined to have a high correlation value of 0.96 in relation to the values obtained by the gas chromatography technique. In a similar work, investigators used an ethyl cellulose QCR coating to selectively detect β-myrcene from mangos, a terpene VOC [257]. By increasing the concentration of their target VOC, investigators confirmed a 0.1 Hz shift of sensor per ppm of terpene. In addition, greater sensitivity was found for β-myrcene relative to other volatile components present in mango samples, such as ocimene, β-carophyllene, carene, and humulene.

Noting the importance of discrimination of mango vapors, and also employing a sensor developed from an earlier investigation [258], another group of researchers compiled a QCM-based sensor array using multivariate analysis for mango crop discrimination with principal component analysis (PCA) [259]. Debabhuti and coworkers employed sensors coated with polydimethylsiloxane (PDMS), β-cyclodextrin, β-mercaptobenzothiazole, polyethyleneglycol 1500, maltodextrin, and gum acacia as QCR coatings to determine the ripening stages of mango samples. Three sources of Indian mangos, Langra, Amrapali, and Himsagar, were obtained and studied, while three stages of ripeness were considered for each crop. Datasets were generated from 24 samples for each crop and employed for PCA analysis. With each crop analysis, the first three principal components accounted for more than 99% of the variance and used to visualize the various stages of ripeness using three-dimensional plots. From resultant PCA graphs, all three mango sources display distinct clusters of unripe, midripe, and ripe harvest samples [259]. In effect, researchers concluded that their QCM-based multi-sensor array was successful in discriminating ripeness and maturity stages among Indian mango crops.

Meat products are commonly known to generate odors after prolonged storage time, and these are a well-known indicator of market freshness. For this reason, Chen and coworkers recently focused on the detection of two VOCs (hexanal and 1-octen-3-ol) from grass carp [260]. Since these VOCs are known to increase over time, these investigators employed a copper(I)-cysteine nanocomposite as a sensing material. Investigators observed that the nanocomposite was unresponsive to changes in humidity and further conducted experiments at 80% relative humidity to replicate water related effects under refrigerated conditions. Comparative studies with other VOCs confirmed selectivity toward the targeted hexanal and 1-octen-3-ol. These investigators employed this sensor using a real sample, exposing vapors from refrigerated grass carp fillets within four storage days. Frequency responses were determined to have a correlation coefficient of 0.96 when compared to results obtained using solid-phase microextraction followed by gas chromatography-mass spectrometry (SPME-GCMS). Although they achieved this high correlation value, the investigators noted that the amplitude of the frequency change within the first two days of exposure did not display the large signal change observed during the last two days of exposure. To further enhance this sensitivity, Chen and coworkers [261] also focused on using graphene oxide as a hydrophobic nanocomposite coating material to detect different aldehydes such as hexanal, octanal, and nonanal, under similar relative humidity conditions in fish samples: grass carp and hairtail fillets. Similar characterizations were performed in fish samples, and investigators further confirmed that increased storage time lead to stable and sufficient responses from their sensor over time. In this work, correlation coefficients from SPME-GCMS of both real samples were determined to be 0.98. While this sensor was not further evaluated in an array using multivariate analysis, the sensitivity of the coating material showed potential use as a suitable material in sensor array applications.

Kalinichenko and Arseniyeva [262] also recently explored the utility of QCM to identify adulterated meat products, such as sausages with different mass percentages of soy protein isolates (SPIs), using QCM coupled with multivariate analysis. By employing seven different polymer-sensing materials, investigators established fingerprint patterns for four different sausage types with varying percentages of SPI adulteration (0, 10, 20, and 30% *w*/*w*). These researchers first employed an algorithm to visualize sensor responses to sausage VOCs in a polygon pattern. This technique provided qualitative information for non-adulterated versus systematically adulterated samples, relaying samples quality via “star” profiles. For qualitative grouping analyses, PCA was employed as an unsupervised method of analysis, while probabilistic neural network (PNN) was used as a supervised method. When the area values (*S_i_*) and maximum value of sensor response (*∆F_max_*) were employed, unsupervised PCA analysis accounted for 96% and 97% of the total variance within the first three principal components, respectively. However, both PCAs were unable to discriminate between 20% and 30% *w*/*w* SPI samples. Investigators then reasoned that a PNN analysis would be best to process this data. Investigators also examined if data training was necessary. After training the raw data with three different methods, via normalizing, auto scaling, and centering, investigators found that the best results for PNN construction were obtained using raw data of *∆F_max_* and *S_i_*, resulting in 100% accuracy and 95.8%, respectively. Ultimately, this work highlights the use of a QCM coupled with multivariate analysis to provide a simple method to assess levels of sausage authentication and introduce novel techniques for food control.

Overall, QCM is a non-destructive, facile, and rapid technique that can be useful in both liquid and gas phase protocols. Thus, from the investigations demonstrated above, QCM may be useful to determine a variety of food quality measures, ranging from adulteration investigations, contamination, and potential harvest queries. As investigations into coating materials, sensor arrays, and exploration with multivariate analyses increases and become available, we hypothesize that this method will become even more useful for analyses in food industries.

## 13. Electrochemical Biosensors for the Detection of Foodborne Pathogens

The detection of foodborne pathogens is a crucial step in the food quality assessment process [263]. Major foodborne illnesses are caused by the contamination of foods with bacteria, such as *Salmonella*, *Clostridium*, *Listeria* and *Vibrio cholerae,* or viruses like norovirus, or protozoans [264]. Electrochemical biosensors have been intensively used to detect such microorganisms in food because of their higher specificity, sensitivity, simplicity, portability, and rapidity in comparison to conventional methods. In this section of the review, we will be discussing recent developments in electrochemical biosensors designed to detect pathogenic microorganisms in food. Figure 1 represents the basic construction and the key components of an electrochemical biosensing strategy specifically designed to detect foodborne pathogens.

In a typical electrochemical biosensor, the analyte is transported towards the working electrode (or the transducer surface) with high mass transport efficiency and it is detected by bio-recognition elements immobilized on the electrode surface which are complementary to the analyte. The analyte concentration is then determined by an electrochemical technique such as voltammetry, amperometry and impedimetry (Figure 1).

Nucleic acids, antibodies, phages, peptides and aptamers are commonly used as biorecognition elements and they play a significant role to determine the specificity of the biosensor. Qian et al. have recently developed a DNA biosensor to detect *Clostridium perfringens* in dairy products [265]. On the other hand, antibodies are considered to be the gold-standard of bio-recognition elements for pathogen detection and such immuno sensors were recently developed to detect *Escherichia coli* O157:H7 [266,267] and *Staphylococcus aureus* in milk samples [268]. Bael et al. [269] have investigated the efficiencies of novel synthetic peptides for electrochemical sensing of human norovirus. In addition, Eissa et al. [270] have constructed a peptide-based electrochemical biosensor for simultaneous detection of two of the most common food-borne pathogens: *Listeria monocytogenes* and *Staphylococcus aureus*. Moreover, aptamers, single-stranded oligonucleotides, are also capable of binding to various food-borne pathogens with high affinity and selectivity. Electrochemical aptasensors have been reported recently for the detection of *Salmonella enterica* [271], *Cronobacter sakazakii* [272], and *Escherichia coli* [273]. Zhou et al. have used T2 bacteriophage (a virus) as the biorecognition element on a screen-printed electrode for rapid and selective detection of *Escherichia coli.*

Nanomaterials have emerged as powerful materials for electrode fabrication due to characteristic features such as high conductivity, surface area to volume ratio, and robustness. Therefore, materials such as nanotubes, nanosheets, and nanoparticles have been extensively utilized to construct biosensor platforms [274,275]. Appaturi et al. have modified a glassy carbon electrode with reduced graphene oxide-carbon nanotube nanocomposites to detect *Salmonella* spp. in food samples [276]. In another recent study, a gold microelectrode was fabricated with single-walled carbon nanotubes for detection of *Yersinia enterocolitica* in fermented vegetable products [277]. Various graphene-based nanocomposites were also reported for the detection of *Salmonella enterica* and *Escherichia coli* [271,278,279]. Moreover, gold nanoparticles have also been extensively incorporated into electrochemical biosensors used in food born pathogen detection, due to their excellent physiochemical properties and biocompatibility [280,281,282,283].

In accordance with the electrical signal output, electrochemical biosensors can be divided into three major categories: voltammetric biosensors, amperometric biosensors, and impedimetric biosensors (Figure 1). In voltammetric techniques such as cyclic voltammetry, differential pulse voltammetry, or square wave voltammetry, the current caused by an electrochemically active redox probe which is directly or indirectly related to the analyte concentration, is measured by varying the potential of the working/sensing electrode. Such voltammetric biosensor platforms have been successfully utilized to detect food-borne pathogens in recent years [282,284]. In amperometric biosensors, the potential of the sensing electrode is maintained at a constant value with respect to a reference electrode and the current is measured as a function time [285]. Amperometric detection of *Escherichia coli* [266,286] and *Samonella* spp. [287,288] in various food samples and *Brettanomyces bruxellensis* [289] in wine products have been investigated. Impedimetry is a label free electrochemical strategy that has become a commonly used technique for pathogen detection in food samples. In contrast to voltammetric and amperometric sensors, the electrochemical impedance is measured when foodborne microorganisms are selectively captured on the sensing electrode surface. The use of impedimetric sensors for the inspection of food quality has been recently reviewed by He and Yuan [290].

## 14. Conclusions and Future Trajectory

This review article reported on the challenges and recent breakthroughs in quality control and the assessment of food products. Overall, recent technological innovations and advances in QCM, electroanalytical devices, real-time polymerase chain reaction methods, DNA barcoding scanners, and NIR spectrometer development have facilitated the accurate, fast, and reliable quality assessment of diverse food products, food raw materials, and ingredients. It has also resulted in effective detection of bacteria, viruses, and other foodborne pathogens in food products. The use of multivariate regression analyses of NIR spectra has incredibly enhanced instrumental calibration, facilitated accurate food sample analyses and reliable quality assessments of various food products. Nonetheless, global quality control and assessment of food products is envisioned to continue to be a global challenge in the coming years for various, complex, and divergent reasons. As a result, the development of effective quality checks, food assessment monitoring strategies, and assessment of food quality schemes by industrial producers, public safety officials, regulatory agencies, and other food stakeholders to ensure global public health and safety is projected to continue to be a top priority. New low-cost instrumentation and technology that will enhance robust, sensitive, and accurate food quality assessment in the field and/or on the production line in the food manufacturing and processing industries are expected to continue to be developed. Many articles are envisioned to report on the use of portable NIR spectrometers for the assessment of food products. Several studies are also anticipated to report on the use of new electroanalytical devices, such as electrode noses and electrode tongues, chemical sensors, multiplex real-time polymerase chain reaction (PCR) techniques, and DNA barcoding scanners for food quality assessment and detection foodborne pathogens in food products. Studies are projected to report on chemometric approaches to instrumental calibration for food quality assessment in the field and/or on the production line in food manufacturing and processing industries. Additionally, numerous articles are expected to report on the use of multivariate regression analysis of NIR spectral data for food analysis and the detection of food adulteration. Emphasis will be placed on the quality control and assessment of processed food shelf-life, evaluation of agricultural produce maturity, and harvest times. Furthermore, quality assessment of edible oils, dairy products, processed foods, and food supplements are projected to be of a considerable research area of interest and a top priority in the coming years. Moreover, several studies are envisioned to report on the quality control and assessment of beverages, meat, pork, fish, eggs, and seafood.

## Figures and Tables

**Figure 1 sensors-20-06982-f001:**
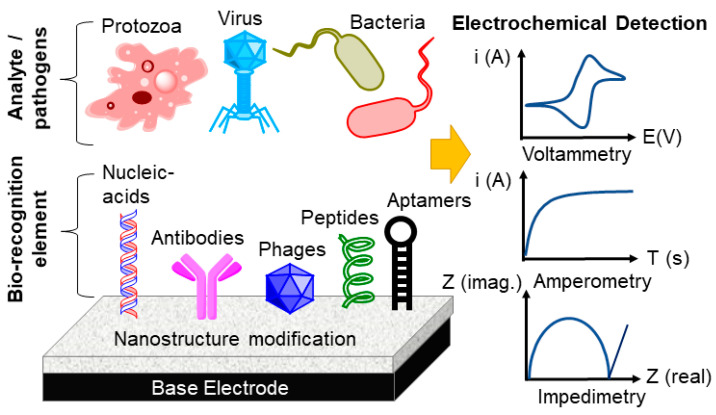
Schematic representation of the key components of a biosensor platform for the detection of foodborne pathogens.

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
