# Peer review of "QCM Sensor Arrays, Electroanalytical Techniques and NIR Spectroscopy Coupled to Multivariate Analysis for Quality Assessment of Food Products, Raw Materials, Ingredients and Foodborne Pathogen Detection: Challenges and Breakthroughs"

_sensors, 2020, doi:10.3390/s20236982_

Round 1

Reviewer 1 Report

Manuscript Title: QCM Sensor Arrays, Electroanalytical Techniques and NIR Spectroscopy Coupled to Multivariate Analysis for Quality Assessment of Food Products, Raw Materials, Ingredients and Foodborne Pathogen Detection: Challenges and Breakthroughs
The current paper reports on the application of several techniques for several tasks in food.
The subject is relevant to the field, and it addresses state of the art methods that are in high demand.
The proposal is sound, and the text is well written and organized. However, the major drawback is that the authors tried to address too many techniques for too many tasks.
For instance, there are several other reviews with comprehensive assessment of only one technique (either NIR spectroscopy, or electrochemical analysis, and so on; for one single product or range of products.
In this sense, the current text is superficial, with little insight and discussion about the subject, and more a list of citations (290 references! It is not even possible to include tables with the references and brief description of the techniques and samples analysed, as it would be huge).
Thus, even with such a large list of papers, a quick search on the subject provides several recent publications that were not addressed.

Hence, in view of the comments above, I can only suggest the authors to at least include some of the references that report the technique for the proposed application, or split the paper to report a comprehensive subject with proper discussion.

Author Response

We are thankful to reviewer 1 for excellent comments about the relevance of our review manuscript to the field. We are also appreciative of reviewer 1’s comments, concerns, and suggestions.  We concur that some review articles are laser focused, addressing one topic that is often tailored for a specific audience. Nonetheless, assessment of food quality spans across many sectors, including instrument manufacturing company, academia, food control officers, food processing industry, and public health. The goal of our review article is to provide a concise literature survey and a summary of general challenges and major breakthroughs in methods of quality checks and assessments of a variety of foods, food raw materials, and ingredients for a wider audience.

We have organized the review article into different sections to allow readers to directly go to sections of interest to them. The summary of the study and the references are provided.  Interested readers may get the copy of the article of interest to them in a more detailed information. Regarding, the missing recent published article, our review article highlights significant published articles between January 2019 and July 2020. It is practically impossible to cite and review thousands of articles published during the review period.

Once again, we are appreciative of reviewer 1 and excellent feedback on our review article.

Reviewer 2 Report

Comments:

Line no 37: in instead of on the processing line.

Lines 40-42: Its would be an overstatement to write this without actual reports to prove this- I suggest to re-word or remove this.

Line 48: NIR needs to be expanded when citing the first time.

Line 61: Change to “introduction and overview”

Line 75- “quality  foods” please explain this term.

Line 79: examines seems to be out of flow, re word as follows: “This review article includes a literature survey and a summary of  general challenges……”

Line 80: “English language publications- Published work in English.”

Line 81: Lines 95-97- Provide a reference for it or term it as a presumption.

Line 99-102: Development of one instrument to measure “quality in general, inclusive of microbiological, adulterants or even presence of hazardous chemicals etc is not practical. Please reword the sentence accordingly. “The development of portable a low-cost instrument that is capable of rapid, reliable, sensitive, 99 accurate, and robust, real-time quality checks, assessments, and assurances of food products in the 100 field and/or at the production line in a food manufacturing or processing industry is still a big 101 challenge.”

Line 113: “emerging contaminants analysis “ change to “Detection of emerging/new contaminants”

Lines 117-118: limited and remains  a significant undertaking- change to limited and requires  a significant undertaking

Line 123-124: “Moreover, a continued decline in government research support hinders 123 creative innovation of new instrumental and technological method development in academia and 124 national labs.” Please provide any references to cite if making such general statements.

Line 143: DNA Barcoding- remove capitals from barcoding.

Line 162: “NIR spectroscopy for quality food assessment is provided”- please reword – assessment of food quality.

Line 163: Remove “through 13”” from Section 171 4 through 13 of the review”

Line 206: Please use pork and not pig meat for consistency.

Lines 210-214: The relevance of lines 209-214 on space science is not clear, would be better to keep the discussion to food matrices.

Line 229: “Importantly, the new reported” please change to “the recently reported.”

Lines 241-243 – please add the reference.

Line 259: Remove the colon

Line 260: “processed foods generated”- add a “has” between foods and generated.

Line 298: Replace “diagnostic” with “diagnostic tool”

Line 306: model.. remove a full stop.

Line 307: Please reword to remove the use of “water “twice.

Line 343: A problem cannot be urgent-it has to be significant or needs urgent attention-please reword accordingly.

Lines 366-370: Add a reference.

Lines 381-384: Please explain that concentrations up to 100% in this study was to train the model-not results for samples.

Lines 458-459: Not all the MS instruments need sample prep or are destructive. Please see below: https://doi.org/10.1007/s00216-013-7316-0, please support your sentence using cost effectiveness and time consumption.

 Line781: The use of NIR in pathogen detection “as per the title” has been completely neglected. Its either necessary to add a section on this or change the title by eliminating pathogens there as no-food borne microbial pathogens have been addressed.

Line 1275: Please specify which food matrices were used as it cannot be general for every food matrix.

Author Response

Reviewer 2: Comments and Suggestions for Authors

We are thankful to reviewer 2 for excellent comments, suggestions, and editorial feedback to improve the quality of our review article. All of the comments and suggestions have been addressed in the revised manuscript.  Please find below specific response to reviewer 2 comments and suggestions highlighted in red below:

Comments:

Line no 37: in instead of on the processing line. This has been changed as suggested.

Lines 40-42: Its would be an overstatement to write this without actual reports to prove this- I suggest to re-word or remove this. We concur with the reviewer. We have removed this statement as suggested by the reviewer.

Line 48: NIR needs to be expanded when citing the first time. NIR has been expanded as suggested.

Line 61: Change to “introduction and overview”: This has been changed as suggested.

Line 75- “quality foods” please explain this term. The suggested changes have been made in the revision to simplify the statement.

Line 79: examines seems to be out of flow, re word as follows: “This review article includes a literature survey and a summary of general challenges……”

Response: We are thankful to reviewer 2. This statement has been changed.

Line 80: “English language publications- Published work in English.”

Response: This statement has been corrected in the revision.

Line 81: Lines 95-97- Provide a reference for it or term it as a presumption.

Response: We have provided relevant references.

Line 99-102: Development of one instrument to measure “quality in general, inclusive of microbiological, adulterants or even presence of hazardous chemicals etc is not practical. Please reword the sentence accordingly. “The development of portable a low-cost instrument that is capable of rapid, reliable, sensitive, 99 accurate, and robust, real-time quality checks, assessments, and assurances of food products in the 100 field and/or at the production line in a food manufacturing or processing industry is still a big 101 challenge.”

Response: We concur with the reviewer and we have made the correction.

Line 113: “emerging contaminants analysis “ change to “Detection of emerging/new contaminants”

Response: This has been changed in the revision as suggested.

Lines 117-118: limited and remains a significant undertaking- change to limited and requires a significant undertaking.

Response: We have made this change in the revision.

Line 123-124: “Moreover, a continued decline in government research support hinders 123 creative innovation of new instrumental and technological method development in academia and 124 national labs.” Please provide any references to cite if making such general statements.

Response: We have provided relevant references

Line 143: DNA Barcoding- remove capitals from barcoding. We have corrected this in the text.

Line 162: “NIR spectroscopy for quality food assessment is provided”- please reword – assessment of food quality.

Response: We have reworded this statement in the revision.

Line 163: Remove “through 13”” from Section 171 4 through 13 of the review”: Response: We have removed “through 13” in the revised manuscript as suggested.

Line 206: Please use pork and not pig meat for consistency.

Response: We have replaced pig meat with pork in the revision.

Lines 210-214: The relevance of lines 209-214 on space science is not clear, would be better to keep the discussion to food matrices.

Response: We thank the reviewer for this excellent comment. However, development of such spectrometers will potentially facilitate the capability and assessment of food quality in space station laboratory in the future. We added that statement in the revision.

Line 229: “Importantly, the new reported” please change to “the recently reported.”

Response: New reported has been changed to recently reported.

Lines 241-243 – please add the reference.

Response: Appropriate reference has been added.

Line 259: Remove the colon:

Response: The colon has been removed.

Line 260: “processed foods generated”- add a “has” between foods and generated. Response: The suggested correction has been made.

Line 298: Replace “diagnostic” with “diagnostic tool”:

Response: We have made the suggested change in the revision.

Line 306: model.. remove a full stop.

Response: The extra period has been removed.

Line 307: Please reword to remove the use of “water “twice.

Response: We have reworded the statement to remove the use of water twice.

Line 343: A problem cannot be urgent-it has to be significant or needs urgent attention-please reword accordingly.

Response: We have reworded the statement accordingly.

Lines 366-370: Add a reference.

Response: The section has been discussed in another section, so it has been removed from line 366-370

Lines 381-384: Please explain that concentrations up to 100% in this study was to train the model-not results for samples.

Response: We concur with the reviewer. We have clarified this statement in the revision.

Lines 458-459: Not all the MS instruments need sample prep or are destructive. Please see below: https://doi.org/10.1007/s00216-013-7316-0, please support your sentence using cost effectiveness and time consumption.

Response: We agree with the reviewer. This statement has been corrected in the revised manuscript.

 Line781: The use of NIR in pathogen detection “as per the title” has been completely neglected. Its either necessary to add a section on this or change the title by eliminating pathogens there as no-food borne microbial pathogens have been addressed.

Response: We are appreciative of this comment. However, the title is appropriate since detection of pathogen were covered in the review.

Line 1275: Please specify which food matrices were used as it cannot be general for every food matrix.

Response: Those studies were conducted in milk samples. We have stated this in the revision.

Reviewer 3 Report

The article fits into the Journal's scope. Subject matter is original and important – food quality control is extremely timeous aspect of research. All research components are present and clearly stated. The abstract is concise and clear. Introduction is well‐written and accurate. Objective of the study is clearly presented. Conclusions are drawn from the analysis of the data collected. References are adequate and based on relevant literature.

The text is adequately written - very, very few editorial and language errors should be corrected (see, for example, lines:  226 (Stanislaw is the author’s first name), 250 (Fabricio is the author’s first name), 306 (additional dot), paragraph 9.0 – line spacing, 859 (additional space)) - but typographical and linguistic errors can be easily fixed.

Author Response

We are thankful to reviewer 3 for the excellent comments and support of our review article. We are also thankful to the reviewer on editorial feedback.  We have addressed the minor grammatical and language errors in the initial submission in the revision. We have also used Stanislaw and Fabricio’s last names in the revision. Finally, the additional dot, line spacing, and space issues have been addressed in the revision.